# Decoding the Epigenome: Comparative Analysis of Uterine Leiomyosarcoma and Leiomyoma

**DOI:** 10.3390/cancers17162610

**Published:** 2025-08-09

**Authors:** Marie Pfaff, Philippos Costa, Haoyu Tang, Bethsebie Sailo, Anup Sharma, Nita Ahuja

**Affiliations:** 1Department of Surgery, Yale School of Medicine, New Haven, CT 06520, USA; marie.pfaff@yale.edu (M.P.); philippos.costa@yale.edu (P.C.); haoyu.tang@yale.edu (H.T.); bethsebie.sailo@yale.edu (B.S.); anup.sharma@yale.edu (A.S.); 2Center for Surgical Science, Department of General, Visceral and Thoracic Surgery, Universitätsklinikum Hamburg Eppendorf, 20246 Hamburg, Germany; 3Office of the Dean, University of Wisconsin School of Medicine and Public Health, Madison, WI 53705, USA

**Keywords:** leiomyoma, leiomyosarcoma, epigenetic, DNA methylation, histone modification, microRNA, non-coding RNA, PI3K/Akt/mTOR pathway, Wnt/β-catenin pathway, TGF-β signaling

## Abstract

Women worldwide face a critical healthcare challenge when uterine smooth muscle tumors are discovered. While most are benign leiomyomas affecting up to 80% of women, a small percentage are deadly leiomyosarcomas with survival rates below 20%. Current diagnostic methods cannot reliably distinguish between these conditions before surgery, leading to devastating consequences: cancer spread through inappropriate surgical techniques or unnecessary fertility-destroying procedures for benign conditions. Understanding the epigenetic differences that drive these tumors could revolutionize women’s healthcare by enabling accurate preoperative diagnosis and personalized treatment approaches.

## 1. Introduction

Uterine leiomyomas (ULM), commonly known as fibroids, represent the most prevalent benign tumors of the female genital tract, affecting approximately 70–80% of women worldwide [1]. Despite their benign nature, these tumors can compromise the quality of life, leading to pelvic discomfort, irregular menstrual bleeding, and infertility [2]. In contrast, uterine leiomyosarcomas (ULMS) are highly aggressive malignant tumors with an incidence of less than 1 per 100,000 women annually [3]. Between these two extremes lies STUMP (Smooth Muscle Tumor of Uncertain Malignant Potential), which fits neither diagnostic category and accounts for approximately 4% of all uterine smooth muscle tumors [4]. This intermediate classification underscores the diagnostic challenges inherent in evaluating uterine smooth muscle neoplasms. ULMS presents a particularly challenging clinical scenario with poor prognosis and high recurrence rates following complete surgical resection. Additionally, these tumors show minimal responsiveness to standard chemotherapeutic interventions. The five-year survival rates dramatically decline in advanced stages, plummeting to approximately 15% [5].

The preoperative diagnosis of ULMS remains one of the most challenging aspects of gynecological oncology, as emphasized by the most recent European Society of Gynecological Oncology (ESGO) guidelines. Currently, no imaging method can definitively exclude sarcoma preoperatively, and misclassification as a benign myoma can lead to inappropriate morcellation, causing systemic spread and transforming a potentially curable disease into a palliative one. Due to these diagnostic uncertainties, the FDA has issued warnings against power morcellation in ULM [6]. While advanced imaging techniques may raise suspicion for sarcoma, definitive preoperative differentiation remains elusive [7,8,9].

Contributing to the complexity of this clinical scenario are ongoing debates about the origin of ULMS. While both conditions originate from the uterine smooth muscle tissue, they represent distinctly different pathological entities with divergent biological drivers. The question of whether ULMS can develop from ULM remains unresolved. Current scientific consensus favors de novo ULMS development, supported by cytogenetic differences and the disparate incidence rates between these entities. However, emerging evidence suggests that a small subset of ULMS may arise from preexisting ULM, particularly in the presence of specific risk factors, as documented in several case reports and molecular studies [10,11,12,13,14]. While genomic analyses have not definitively resolved this controversy, epigenetic investigations may provide new insights into this complex pathogenic relationship.

These epigenetic approaches have become increasingly important as traditional genomic studies reach their limitations in distinguishing these tumor types. Emerging evidence has already shown that ULM and ULMS share both overlapping and distinct epigenomic features, which hold promise as biomarkers for differential diagnosis and development of targeted therapies [15,16]. Epigenetic mechanisms—including DNA methylation, histone modifications, and non-coding RNAs—have emerged as crucial regulators in both tumor types. DNA methylation involves the addition of a methyl group to cytosine in CpG dinucleotides, catalyzed by DNA methyltransferases (DNMT). DNMT1 maintains existing marks, while DNMT3A and DNMT3B establish new ones (de novo) [17]. Demethylation is initiated by Ten-Eleven Translocation (TET) enzymes, which oxidize 5-methylcytosine (5-mC) into intermediates that lead to methyl group removal [15]. Histone modifications are post-translational modifications that regulate gene expression by altering chromatin structure and accessibility. These modifications include acetylation, methylation, phosphorylation, and ubiquitination. MicroRNAs (miRNAs) are small, non-coding RNAs of approximately 22 nucleotides that regulate gene expression post-transcriptionally by binding to the 3′ untranslated region (UTR) of target mRNAs, typically leading to translational repression or mRNA degradation [18]. Long non-coding RNAs (lncRNAs) are longer than 200 nucleotides and can regulate gene expression through various mechanisms, including chromatin modification, transcriptional regulation, and post-transcriptional processing. Understanding these epigenetic mechanisms is therefore critical, as they may hold the key to resolving current diagnostic challenges. Given the profound implications for treatment and prognosis, there is an urgent need for reliable molecular biomarkers that can accurately distinguish between these entities when conventional approaches fail.

This narrative review synthesizes current evidence on epigenetic similarities and differences between ULM and ULMS to identify diagnostic biomarkers and therapeutic targets for improved clinical management. We also consider what current epigenetic data may contribute to understanding the ULM-ULMS relationship.

## 2. Epigenetic Alterations

### 2.1. DNA Methylation Landscapes

DNA methylation patterns provide valuable insights into the molecular pathogenesis of uterine smooth muscle tumors and may serve as potential biomarkers for differential diagnosis. Global DNA hypomethylation in combination with selective gene hypermethylation, especially of tumor suppressor genes, has been uncovered in different cancers [19,20,21,22]. Current evidence reveals distinct methylation patterns between ULM and ULMS, though most findings come from small cohort studies, which limit definitive conclusions.

ULM and ULMS exhibit fundamentally opposite methylation enzyme profiles that reflect their biological differences. ULM shows global hypomethylation compared to normal myometrium, with significantly elevated 5-hydroxymethylcytosine (5-hmC) levels and upregulated TET1/TET3 demethylating enzymes. This contrasts with the decreased 5-hmC levels typically found in malignant tumors [17,23,24]. DNMT enzyme expression, however, exhibits inconsistent patterns in ULM across studies, with variable DNMT1 levels and generally decreased DNMT3A/3B expression [25,26]. Conversely, ULMS demonstrate elevated DNMT1 and DNMT3A methylating enzymes, though their specific genomic targets remain unclear [3,27]. This reciprocal enzyme expression pattern provides a mechanistic basis for their distinct methylation landscapes.

While DNA methylation-based hierarchical clustering groups ULM and ULMS together were comparable to other uterine tumor types, direct comparison reveals distinct methylation signatures that distinguish these entities [3,28].

However, evidence for ULMS-specific global methylation changes compared to ULM remains preliminary and contradictory, derived from only two small studies with fundamental methodological differences. While both studies demonstrate global ULMS hypomethylation patterns, their findings regarding genome-wide methylation distribution diverge substantially, highlighting the nascent state of this research field. Miyata et al. found that ULMS hypomethylation primarily affects intergenic regions and gene bodies, while CpG islands were paradoxically more hypermethylated, whereas Conconi et al. reported reduced CpG island methylation in aggressive cases. The contradictory findings primarily reflect methodological differences rather than biological variation. Miyata et al. employed genome-wide methylation analysis across three ULMS, three ULM, and three normal myometrium tissue samples (n = 9 total) [29]. In contrast, Conconi et al. used CpG island-focused analysis across 3 ULM, 14 STUMP, and 5 ULMS samples, with STUMPs comprising 64% of their cohort [30]. The substantial differences in genomic coverage and the inclusion of diagnostically uncertain STUMP cases likely explain their conflicting results.

The most clinically relevant distinction emerges from hormone-methylation interactions, which provide a molecular explanation for differential hormone responsiveness between ULM and ULMS. In ULM, a complex bidirectional network exists where hormone signaling both influences and is influenced by methylation patterns. While the majority of ULM cells express hormone receptors and remain hormone-responsive, leiomyoma stem cells (LSCs) show reduced *TET1/TET3* expression. This results in dual-level methylation silencing of both the progesterone receptor (*PR*) gene locus and its target genes, directly inhibiting PR expression and suppressing progesterone-induced differentiation pathways [31]. This methylation-mediated silencing explains why LSCs require paracrine signaling from hormone-responsive cells for proliferation and accounts for the variable efficacy of hormone-based therapies in ULM treatment. In contrast, hormone-responsive ULM cells show the opposite pattern. PR signaling upregulates TET enzymes, creating a feedback loop where hormone activation promotes global demethylation. Somatic *MED12* mutations, present in ~70% of ULMs, enhance this hormone-methylation network by increasing PR target binding [32]. Additionally, PR maintains estrogen receptor alpha (ERα) expression through promoter demethylation in hormone-responsive cells, while both hormone receptors are silenced by hypermethylation in LSC [33]. The therapeutic relevance of this mechanism is demonstrated by the finding that demethylating agents can restore hormone receptor expression and therefore hormone sensitivity in LSC, making combination approaches with antihormonal drugs particularly promising [31,33].

ULMS demonstrates a fundamentally different pattern where hormone receptor hypermethylation affects the entire tumor mass rather than a small stem cell population. Hasan et al. found progesterone receptor DNA hypermethylation throughout ULMS compared to extrauterine leiomyosarcomas, contrasting with ULM’s predominantly high hormone receptor expression [34]. This genome-wide methylation-mediated silencing of hormone receptors provides a molecular explanation for why ULMS typically lacks hormone dependence and shows poor response to hormone-based treatments. Whether demethylating agents can similarly restore hormone sensitivity in ULMS, as demonstrated in ULM LSC, remains to be investigated.

Beyond these global patterns, individual methylated genes have been investigated in both tumor entities and are listed in Table 1. Current methylation evidence suffers from critical limitations: extremely small sample sizes (3–5 ULMS cases per study), contradictory findings regarding key patterns, and substantial methodological heterogeneity. A fundamental issue is that ULMS are frequently investigated as a subgroup within heterogeneous sarcoma studies rather than as a distinct uterine entity. The field urgently needs large, standardized methylation studies comparing ULMS directly to ULM. Current evidence is too preliminary for clinical application. Future studies should employ prospective, multi-center designs with adequate sample sizes (≥50 cases per group) of both ULM and ULMS, standardized high-resolution methylation platforms, centralized pathology review, and integrated analysis of methylation patterns with clinical outcomes to develop reliable diagnostic and prognostic biomarkers [35].

### 2.2. Histone Modifications and Chromatin Remodeling Mechanisms

DNA is organized into chromatin by wrapping around histone proteins, forming nucleosomes—the basic units of chromatin structure. The tails of these histones undergo various post-translational modifications that influence how tightly DNA is packed, thereby regulating gene accessibility and transcription. The most studied histone modifications are methylation and acetylation [35]. These marks are added by writers, interpreted by readers, and removed by erasers. For instance, EZH2, a methyltransferase within the polycomb repressive complex 2 (PRC2), deposits H3K27me3, a repressive mark that silences gene expression. In contrast, H3K4me3 is associated with active transcription [39,40]. Histone acetylation similarly promotes gene expression by loosening chromatin structure, and is reversed by histone deacetylases (HDAC), which tighten chromatin and repress transcription.

In addition to chemical modifications, chromatin accessibility is further regulated by chromatin remodeling complexes, which use ATP to reposition or replace nucleosomes. The SWI/SNF complex, for example, facilitates nucleosome sliding and includes BRD9 (Bromodomain containing 9), a reader protein that binds acetylated histones to influence transcription, and *ATRX* (α-thalassemia/intellectual disability X-linked gene), which maintains chromatin structure and genomic stability. These layers of histone modifications and chromatin remodeling work together to fine-tune gene expression in both normal and disease states [41,42,43].

In ULM, histone methylation plays a key role in regulating genes involved in tumorigenesis [26,40]. Globally, ULM exhibits reduced H3K4me3—an activating histone mark—and increased H3K27me3, which is associated with transcriptional silencing [39,40]. This shift could be driven by EZH2 overexpression, with one study showing approximately 70% of ULMs showing elevated EZH2 levels compared to normal myometrium [44]. In contrast, another study found no EZH2 expression, likely due to technical differences [45]. However, these methylation changes are not uniform across all genes. H3K4me3 is selectively enriched at the promoters of proto-oncogenes such as *SATB2*, *DCX*, *SHOX2*, *ST8SIA2*, *CAPN6*, and *NPTX2*, while it is depleted at tumor suppressors like *KRT19*, *ABCA8*, and *HOXB4* [39]. Similarly, *MSH2*, a tumor suppressor involved in DNA mismatch repair, is altered in up to 90% of ULM, and its expression inversely correlates with H3K27me3 enrichment at its promoter. Experimental data show that EZH2 overexpression suppresses *MSH2*, while EZH2 inhibition restores its expression, reinforcing EZH2’s gene-specific silencing role [40].

ULMS and ULM exhibit fundamentally distinct patterns of histone acetylation, which may contribute to their divergent biological behavior. ULMS shows increased expression of histone acetyltransferase HAT1 and Class I histone deacetylases (HDAC1, HDAC2, HDAC3) compared to ULM and normal myometrium, suggesting a disruption in the balance between histone acetylation and deacetylation [3,46]. HDAC6, a primarily cytoplasmic deacetylase, is more strongly expressed in ULM than in normal myometrium and has been functionally linked to ERα signaling. Its silencing leads to reduced ERα protein levels [15,47]. These imbalances present therapeutic opportunities, as Class I HDAC inhibition with Tucidinostat induces antiproliferative effects in ULMS cell lines by altering histone modification patterns and modulating cell cycle gene expression [46].

BRD9 has been implicated in both ULM and ULMS, with significantly higher expression in the latter [48,49]. In ULM, BRD9 inhibition induces G1 cell cycle arrest and greater growth suppression than in normal uterine smooth muscle cells, accompanied by upregulation of *CDKN1A*, *DNMT3A*, *TET1*, and *TET2*, and downregulation of *PCNA* and *EZH2* [50]. In ULMS, targeted BRD9 inhibition with TP-472 leads to apoptosis and cell cycle arrest, along with broad transcriptional changes affecting PI3K/Akt, inflammatory response pathways, and other regulatory mechanisms [51]. Additionally, tumors with SRCAP alterations exhibit reduced levels of the histone variant H2A.Z at promoter regions, thereby impairing chromatin accessibility and disrupting the expression of differentiation-related genes [52].

One of the most striking molecular distinctions between ULM and ULMS involves alterations in *ATRX*. *ATRX* is frequently mutated or lost in ULMS (16–31%), particularly in poorly differentiated or undifferentiated tumors, which are associated with more aggressive histologic features and worse clinical outcomes [52,53,54,55]. Loss of ATRX triggers the alternative lengthening of telomere (ALT) phenotype, a telomerase-independent mechanism of telomere maintenance. However, loss of ATRX is not the sole driver of the ALT phenotype in ULMS, as the prevalence of ALT activity (53–78%) significantly exceeds the frequency of *ATRX* alterations and is markedly higher than in most cancers [56]. ULM, in contrast, rarely exhibits *ATRX* alterations, and the ALT mechanism is virtually absent in these benign tumors, which maintain normal telomere length without activating ALT pathways. In rare instances when *ATRX* loss occurs in ULM, it is associated with a more aggressive phenotype, potentially predisposing to pulmonary benign metastasis [57]. These findings establish ATRX as a promising biomarker for distinguishing ULM from ULMS and as a potential therapeutic target, particularly given the vulnerability of ATRX-deficient cells to DNA damage-inducing agents.

### 2.3. Non-Coding RNA Networks

MiRNAs and lncRNAs represent a promising frontier in uterine smooth muscle tumor research. Despite limited data on ncRNA expression in ULMS, these regulatory molecules offer substantial potential as therapeutic targets and diagnostic biomarkers. Liquid biopsy technologies enable the development of circulating miRNA-based assays for minimally invasive early detection and differentiation of uterine mesenchymal malignancies [58,59,60,61]. Comparative ncRNA profiling serves a dual purpose: divergent signatures may facilitate non-invasive diagnostics, while conserved patterns could reveal shared oncogenic mechanisms and therapeutic vulnerabilities across the spectrum of smooth muscle tumors.

#### 2.3.1. ULM-Specific miRNA Patterns

Several studies have characterized aberrant miRNA expression in ULM compared to normal myometrium, revealing distinct functional pathways affected by epigenetic dysregulation.

The miR-29 family plays a central role in ECM regulation and is downregulated in ULM, with miR-29b and miR-29c showing particular significance. MiR-29 family members regulate ECM genes by reducing collagen expression when overexpressed and promoting tumorigenesis when suppressed [62,63]. When expressed at normal levels, miR-29c functions as a negative regulator of TGF-β3. Its downregulation in ULM tissues leads to enhanced TGF-β3 expression and signaling activity [64]. Notably, overexpression of miR-29 has been linked to accelerated aging, suggesting its downregulation may be essential for tumor maintenance [65].

Multiple miRNAs affecting hormone signaling have been identified in ULM. MiR-27b, significantly downregulated in ULM compared to normal myometrium, inversely regulates the estrogen metabolism enzyme CYP1B1 [66,67]. Similarly, miR-206 is downregulated in ULM, normally binding to ERα, reducing its expression through post-transcriptional regulation. It has also been proposed as a differentiating biomarker for uterine smooth muscle tumors [68]. The hormonal regulation extends to the miR-29 family, as estrogen and progesterone suppress miR-29b and promote collagen overexpression, which is particularly significant given that ULM is a primarily hormone-driven tumor. This hormonal axis may partially explain the hormone dependency of ULM, highlighting the critical interplay between hormonal influence and epigenetic aberrations [63,69].

The miRNA landscape in ULM also encompasses critical cell survival and proliferation pathways. For instance, ULM tissue exhibits lower levels of miR-93 compared to normal myometrium, which contrasts sharply with malignant tumors like breast cancer where miR-93 is upregulated and promotes cancer progression by targeting *PTEN* and activating the PI3K/Akt pathway [18,70,71]. In ULM, miR-93 displays an inverse relationship with IL-8, with IL-8 being overexpressed in ULM tissue [69]. Experimental upregulation of miR-93 in ULM cells leads to reduced cellular viability, inhibited cell cycle progression, and enhanced apoptosis [72]. This highlights the complexity of miRNA function in benign versus malignant tumor development.

Several other miRNAs have been identified in ULM with altered expression patterns, though most of these findings have not been consistently reproduced across publications [73].

#### 2.3.2. Long Non-Coding RNA in ULM

LncRNAs provide important regulatory functions in ULM development. H19, the most thoroughly investigated lncRNA, is aberrantly upregulated in ULM and functions as an oncogene most likely via the Wnt/β-catenin and PI3K/Akt/mTOR pathways. Mechanistically, H19 acts as a molecular sponge for let-7 miRNA, consequently upregulating the expression of *HMGA2* and *TET3* [68,74]. Since TET enzymes are upregulated in ULM, this amplifies the epigenetic alterations characteristic of these tumors [75]. H19 also functions as an estrogen receptor (ER) modulator, with plasma levels correlating with ER and PR expression and predicting postoperative recurrence [76].

Interactions between lncRNAs and the hormonal axis are frequent. XIST, a lncRNA critical for X-chromosome inactivation, is involved in hormone-dependent proliferation of ULM, activation of the Wnt signaling pathway, and collagen gene expression. XIST can function as both an oncogene and tumor suppressor and is proposed to act as a sponge for miR-29c and miR-200c, thereby increasing collagen gene expression [77,78]. The negative correlation between XIST and miR-29c levels supports this sponge mechanism, linking hormone-induced miR-29 suppression to enhanced TGF-β3 signaling and ECM accumulation [26,79,80,81]. This highlights a potential signaling pathway where hormone-dependent ULM proliferation involves XIST upregulation, miR-29c downregulation, increased ECM production, and TGF-β signaling. Lentiviral-mediated XIST knockdown in xenograft models significantly reduced tumor growth, supporting the hormone/XIST/miR-29 interaction as a potential therapeutic target [82]. Additional lncRNAs exhibit similar regulatory mechanisms. LncRNA MIAT, upregulated in ULM, exhibits a comparable miR-29 sponge effect to XIST, and its knockdown also leads to reduced tumor growth in vivo [80,83].

LncRNA APTR is also upregulated and promotes ULM proliferation in an ER-dependent manner. ER knockdown abolishes this proliferative effect by lncRNA APTR [84].

Another noteworthy lncRNA is LINCMD1, which is decreased in ULM tissue and normally functions as a molecular sponge for miR-135b. Its absence increases miR-135b, which decreases *APC* and activates the Wnt/β-catenin pathway, thereby promoting collagen accumulation [85].

LncRNA expression differs between ULM subtypes. LncRNA AL445665.1-4 directly binds and regulates miR-146b, which shows opposing expression between ULM subtypes, being downregulated in solitary but upregulated in multiple ULMs [86].

Many other lncRNAs are dysregulated in ULM, including the upregulated CAR Intergenic 10, whose knockdown induces cell cycle arrest through mechanisms that require further investigation [87,88].

Collectively, these lncRNAs establish a complex regulatory network in ULM, predominantly functioning as miRNA sponges that modulate hormone-responsive pathways, particularly involving miR-29 family members and TGF-β signaling, ultimately promoting ECM accumulation and tumor growth.

#### 2.3.3. ULMS-Specific miRNA Patterns

ULMS demonstrate distinctive miRNA expression profiles that differ from both normal myometrium and ULM, reflecting their malignant phenotype. However, while there are studies describing divergent miRNA expressions, their functions in ULMS specifically are under-investigated and mostly only looked at in other sarcoma subtypes, highlighting the need for more extensive ULMS-specific studies. The limited number of ULMS-specific functional studies represents a significant knowledge gap that restricts mechanistic understanding and therapeutic target validation.

MiR-221 is upregulated in ULMS compared to ULM, where it is either undetectable or expressed at lower levels than in normal myometrium, marking an important difference [89,90]. While its direct mechanistic role in ULMS pathogenesis remains to be elucidated, findings from osteosarcoma research suggest it enhances cell proliferation, invasion, and migration, potentially applicable to ULMS given similar malignant characteristics [67]. Due to the observed correlation between its expression and osteosarcoma severity, researchers have proposed miR-221 as a potential biomarker for osteosarcoma, although investigations to date have been limited to tissue samples [91,92]. This biomarker could potentially also be evaluated for differential diagnosis between ULM and ULMS.

Notably, miR-152 is significant due to its involvement in the PI3K/Akt/mTOR pathway. This miRNA is downregulated in ULMS, leading to increased expression of *MET* and *KIT*, which are both activators of the PI3K/Akt/mTOR pathway. Reduced miR-152 expression is associated with increased aggressiveness in ULMS, with experimental upregulation leading to cell death [93,94,95].

Several circulating miRNAs represent promising biomarker candidates for ULMS detection and monitoring. MiRNAs also appear to influence response to therapy in ULMS. For example, miR-34a, miR-17, and miR-106a have been associated with non-response to eribulin treatment [96]. Understanding these associations may help guide therapeutic decisions and develop strategies to overcome treatment resistance.

#### 2.3.4. Long Non-Coding RNA in ULMS

Research on lncRNAs in ULMS remains limited, with most studies encompassing broader soft tissue sarcoma (STS) cohorts that include multiple histological subtypes. This heterogeneity poses challenges in identifying ULMS-specific lncRNA signatures and mechanisms.

Among the few lncRNAs characterized specifically in leiomyosarcoma (LMS), EGFR-AS1 demonstrates significant clinical relevance. This lncRNA is upregulated in LMS tissues and cell lines and negatively correlates with CD8+ T-cell infiltration. EGFR-AS1 enhances LMS-CD8+ T-cell interactions and promotes CD8+ T-cell apoptosis through the PD-1/PD-L1 checkpoint pathway, contributing to immune evasion in LMS cells [97].

A 5-lncRNA prognostic signature (AC018645.2, LINC02454, ERICD, DSCR9, AL031770.1) was developed for STS and showed superior predictive accuracy specifically in LMS patients compared to other STS subtypes, though no individual lncRNA emerged as particularly of interest for ULMS [98].

The current research landscape highlights the urgent need for ULMS-specific lncRNA studies to better understand their therapeutic potential and develop targeted treatment strategies for this aggressive malignancy.

#### 2.3.5. Shared ncRNA Patterns Between ULM and ULMS

While lncRNA research in ULMS remains limited, preventing meaningful comparison with ULM, several studies have identified shared miRNA signatures between these tumor types. These similarities may reveal common oncogenic mechanisms, though expression level differences between ULM and ULMS could still serve diagnostic purposes.

The let-7 and miR-200 families show striking similarities with downregulation in both ULM and ULMS dysregulation. Let-7 generally promotes terminal differentiation, inhibits cell cycle reentry, and is involved in stem cell differentiation as well as cell cycle regulation, with an inverse correlation between Ki67 and let-7 expression [67,99,100]. In malignancies such as lung cancer, upregulation of let-7 in mouse models has been shown to reduce tumor growth [101]. In ULM, in vitro functional miRNA studies showed repression of *HMGA2* by let-7 miRNA. Since *HMGA2* is one of the key genomic mutations in the development of ULM, though not consistently mutated, the interaction with let-7 could represent an alternative mechanism to interfere with *HMGA2* function. Expression of *HMGA2* and let-7 correlates with ULM size, with larger ULMs showing higher HMGA2 levels and lower let-7c levels than smaller ULM [18,102]. While ULMS also exhibit global downregulation of the let-7 family, they demonstrate different functional correlations compared to ULM, notably lacking the tumor size correlation observed in ULM. Decreased expression of specific members correlates with worse overall survival, higher metastasis rates, and poorer disease-free survival, respectively [89]. The let-7 family is proposed to function as a tumor suppressor in this context and demonstrates similar mechanisms in both tumor types. This inverse relationship between let-7 expression and tumor size in ULM, contrasted with the consistently low expression in ULMS, suggests that progressive downregulation of let-7 might contribute to more aggressive tumor growth.

The miR-200 family shows significant downregulation in both tumor types, with ULMS exhibiting more pronounced reduction than ULM and other sarcoma subtypes [89,103]. At the molecular level, miR-200 regulates EMT through an inverse correlation with *ZEB1* and *ZEB2*, transcription factors that drive this transition [69]. Experimental overexpression of miR-200c reduces ZEB1/ZEB2 while increasing E-cadherin expression. Beyond affecting cellular phenotype, miR-200 also influences ECM composition [18,104]. Low miR-200c levels result in increased IL-8 expression with anti-apoptotic effects, a characteristic feature observed in ULM [105]. Functional studies demonstrate that overexpressing miR-200c and miR-200a in ULM cells reduces proliferation and transforms cells from a fibroblastoid to an epithelial phenotype [18]. Similarly, in ULMS cell lines, miR-200c overexpression increases caspase 3/7 activity while inhibiting proliferation and migration, though these mechanisms remain less investigated than in ULM [89,103].

MiR-1 has been shown to be downregulated in both ULM and ULMS compared to normal myometrium. While there is not much known about its function in ULMS, miR-1 is also significantly downregulated in rhabdomyosarcoma and osteosarcoma tissue samples [93]. Reintroduction of miR-1 in rhabdomyosarcoma and osteosarcoma cell lines has suggested a tumor suppressor effect [106,107]. In breast cancer, higher serum miR-1 levels have been shown to correlate with a better response to neoadjuvant chemotherapy [108]. While the functional role of miR-1 is unclear, there is potential for miRNA therapeutics targeting both ULM and ULMS.

The parallel dysregulation of both let-7 and miR-200 families, as well as miR-1, suggests potential shared therapeutic targets for ULM and ULMS, despite their different malignant potential.

The limited research on lncRNAs in ULMS compared to ULM precludes meaningful comparison of lncRNA profiles between these tumor types. This knowledge gap underscores the critical need for comprehensive lncRNA profiling studies specifically focused on ULMS to elucidate potential similarities and differences with ULM.

#### 2.3.6. Divergent ncRNA Patterns in ULM and ULMS

Distinctive miRNA expression patterns between ULM and ULMS offer significant potential for differential diagnosis and risk stratification.

MiR-221, as previously mentioned, shows opposing expression patterns in ULMS and ULM and has been proposed as a biomarker in other STS, showing great potential for future investigations. MiR-206 is also differentially expressed and connected with the hormonal axis, which could explain the different responses to antihormonal treatment of both entities.

Several studies have identified promising miRNA biomarker panels for clinical applications. Yokoi et al. investigated both serum and histological samples from patients with various uterine tumors and analyzed their miRNA profiles [109]. Their research identified a diagnostic panel consisting of miR-1246 and miR-191-5p capable of distinguishing between ULMS and ULM. MiR-1246 has been investigated as a potential biomarker for various malignancies, being most prominent in breast and lung cancer [58,110]. Additionally, miR-144-3p, miR-34a-5p, and miR-206 have demonstrated utility in distinguishing ULMS from benign uterine smooth muscle tumors, although these findings are currently limited to tissue samples and have yet to be transferred to liquid biopsies [111]. MiR-34a-5p is hereby upregulated in both ULM and ULMS compared to normal myometrium, but significantly more upregulated in ULMS compared to ULM.

Combining radiological findings with liquid biopsies analyzing miRNA profiles could enable improved risk stratification prior to intervention, potentially addressing both delayed diagnosis of ULMS and overly radical interventions for benign ULM. Key miRNA expression profiles and their clinical implications are summarized in Table 2.

## 3. Convergent Signaling Pathway Dysregulations

ULM and ULMS appear to converge on the same dysregulated molecular pathways, despite arising from different genomic and epigenomic alterations. Some common molecular pathways dysregulated in both ULM and ULMS are the Wnt/β–catenin, PI3K/Akt/mTOR, and TGF-β signaling. Understanding how diverse upstream epigenetic and genetic events ultimately dysregulate the same signaling axes opens the door to unified therapeutic strategies.

### 3.1. Wnt/β-Catenin Pathway

The Wnt/β-catenin pathway is critical for embryonic development, tissue homeostasis, and stem cell regulation. Its dysregulation is implicated in various malignant and benign tumors, such as colorectal adenocarcinoma (CRC) and desmoid tumors [113,117,118]. Activation of this pathway happens when Wnt ligands bind to Frizzled/low-density lipoprotein receptor-related protein (LRP)5/6 receptors, inhibiting the β-catenin destruction complex, consisting of Axin, APC, glycogen synthase kinase 3β (GSK3β), and casein kinase 1 (CK1). This prevents β-catenin degradation, allowing it to accumulate, translocate to the nucleus, and drive T-cell factor/lymphoid enhancer-binding factor (TCF/LEF)-mediated transcription [114]. In ULM and ULMS, genomic, epigenomic, and protein expression abnormalities interact with this pathway. While both ULM and ULMS demonstrate activation of this central pathway, they achieve this activation through fundamentally distinct molecular mechanisms that reflect their different pathobiologies. Figure 1 provides a comprehensive overview of Wnt/β-catenin pathway dysregulation, displaying the central signaling cascade and its distinct regulatory mechanisms in ULM versus ULMS. The left side shows ULM-specific alterations (blue boxes), while the right side depicts ULMS-specific changes (red boxes). ULM achieves pathway activation through intracellular mechanisms, while ULMS manipulates receptor–ligand signaling balance to achieve the same downstream transcriptional outcome.

ULM employs primarily intracellular regulatory defects affecting transcriptional, chromatin, and RNA-mediated mechanisms to dysregulate Wnt/β-catenin signaling (Figure 1 (I)). Genomically, 70% of ULMs carry a gain-of-function (GOF) *MED12* mutation [115]. Knockout of *MED12* leads to downregulation of Wnt/β-catenin signaling, while introducing a GOF mutation leads to its upregulation, likely mediated by inhibition of GSK3β [119,120]. This represents a direct disruption of intracellular transcriptional machinery. Epigenomically, ULM shows global H3K4me3 reduction and downregulation of LINCMD1 (Long intergenic non-protein coding RNA muscle differentiation 1) [39,85]. H3K4me3 loss activates *SATB2* and *SHOX2*, which in turn promote Wnt/β-catenin signaling [39]. While the precise mechanisms remain unclear, high *SHOX2* expression correlates with *RUNX2* upregulation—an established Wnt/β-catenin activator—and SATB2 alters chromatin structure via matrix-attachment region binding, facilitating transcription of Wnt/β-catenin pathway genes [121,122].

Reduced LINCMD1 increases miR-135b, which targets *APC* and destabilizes the β-catenin destruction complex [85]. These chromatin-level and RNA-regulatory modifications create a permissive intracellular environment for sustained pathway activation. Wnt5b is overexpressed in ULM, and hormonal factors induce Wnt4/11/16 expression, all ligands and activators of Wnt/β-catenin signaling [123]. Thus, ULM achieves Wnt activation through a multi-layered approach combining genetic mutations, epigenetic modifications, RNA dysregulation, and ligand overexpression.

In stark contrast, ULMS predominantly dysregulates Wnt/β-catenin signaling through altered receptor–ligand signaling balance by modifying the expression of pathway modulators and receptors. In ULMS, the Wnt/β-catenin pathway appears upregulated in a subset of patients, driven by epigenetic factors (e.g., secreted frizzled-related protein 4 (*SFRP4*) silencing) and possibly modulated by non-canonical Wnt signaling (e.g., receptor tyrosine kinase-like orphan receptor 1 (ROR1)) (Figure 1 (II)). Reported rates of pathway activation vary widely (6–64%), likely reflecting both biological heterogeneity and methodological differences [124,125,126]. Rather than disrupting core intracellular machinery like ULM, ULMS manipulates the receptor–ligand signaling environment. Epigenetic silencing of *SFRP4*, a Wnt pathway antagonist, and elevated expression of frizzled class receptor 6 (FZD6) have been proposed as potential mechanisms driving pathway activation [127,128,129,130]. This creates a scenario where pathway activation occurs through elimination of extracellular inhibitory signals and amplification of receptor availability rather than through internal regulatory dysfunction. The non-canonical Wnt pathway may also influence ULMS. ROR1 is a receptor for non-canonical Wnt ligands such as Wnt5a [131]. Binding of Wnt5a to ROR1 activates non-canonical Wnt signaling pathways, which antagonize canonical Wnt/β-catenin signaling. This inhibition is thought to occur through sequestration of shared Wnt ligands or disruption of canonical Wnt receptor complex formation [132]. Their low expression in ULMS suggests reduced non-canonical activity, possibly shifting the balance toward β-catenin-driven transcription [133]. This represents a sophisticated manipulation of receptor balance rather than direct disruption of intracellular pathway components.

These mechanistic differences have profound implications for therapeutic targeting. While both tumor types ultimately activate the same downstream Wnt/β-catenin transcriptional programs, the distinct upstream mechanisms suggest that ULM might be more susceptible to therapies targeting intracellular regulatory machinery, such as chromatin-modifying agents and *MED12*-targeted therapies, whereas ULMS might respond better to strategies targeting receptor-ligand balance, such as SFRP4 restoration or FZD receptor inhibition. This mechanistic divergence leading to pathway convergence exemplifies how different tumors can exploit the same oncogenic pathway through entirely different molecular strategies.

### 3.2. PI3K/AKT/mTOR Pathway

The PI3K/AKT/mTOR signaling pathway plays a critical role in uterine tumor biology, with immunohistochemical analyses demonstrating that nuclear phosphorylated Akt and mTOR expression levels increase progressively from benign ULM to intermediate STUMP to malignant ULMS [10]. This establishes a correlation between pathway activation and increasing malignant potential.

#### 3.2.1. Pathway Mechanism and Regulatory Network

When ligands bind to receptor tyrosine kinases (RTKs) or insulin-like growth factor 1 receptor (IGF1R), they activate PI3K, which phosphorylates PIP2 to PIP3. This recruits Akt for partial activation, with full activation requiring mechanistic target of rapamycin complex 2 (mTORC2). Once fully activated, Akt stimulates mTORC1, which regulates cellular growth. Simultaneously, Akt inhibits several key proteins: GSK3α/β (linking to Wnt/β-catenin pathway), forkhead box O (FOXO) proteins (blocking apoptosis), and p53 (via mouse double minute 2 homolog (MDM2) phosphorylation) [134]. Aberrant PI3K/Akt/mTOR pathway activation promotes proliferation and inhibits apoptosis. It also reprograms cellular metabolism and facilitates EMT. Additionally, it modulates immune responses. Collectively, these effects support tumor development, cancer progression, and therapy resistance. In ULMS, hyperactivation of the PI3K/Akt/mTOR signaling pathway is observed in 30–33% of cases and serves as a critical driver of pathogenesis [3,135]. This pathway creates a complex regulatory network through bidirectional interactions with epigenomic and genomic mechanisms, becoming activated through them while simultaneously influencing them. Figure 2 provides a comprehensive overview of this pathway network, displaying the central PI3K/Akt/mTOR signaling cascade and its multiple regulatory inputs and outputs. The left side shows ULM-specific alterations (blue boxes), while the right side depicts ULMS-specific changes (red boxes). Interactions with TGF-beta signaling are also shown and discussed in the next chapter.

#### 3.2.2. Genomic Alterations

Nuclear receptor co-repressor 1 (NCOR1) amplification represents a key genomic alteration affecting pathway regulation. NCOR1 is a transcriptional corepressor that is frequently amplified in ULMS (10% of cases) and whose nuclear localization and activity depend on PI3K/Akt-mediated control (Figure 2, IIa). Since HDAC3’s deacetylase activity is dependent on activated NCOR1, targeting the PI3K/Akt/mTOR pathway represents a promising therapeutic strategy that could simultaneously disrupt multiple epigenetic oncogenic processes in ULMS by interfering with NCOR1 function and downstream HDAC3 activity [135,136].

*PTEN* mutations constitute another major genomic alteration in ULMS. Genomically, PTEN, a key negative regulator that dephosphorylates PIP3 and consequently inhibits PI3K/Akt/mTOR signaling, frequently exhibits mutational inactivation or deletion in ULMS (Figure 2, IIb). Comparative genomic analyses demonstrate that PTEN aberrations and subsequent pathway activation occur at significantly higher frequencies in LMS compared to other sarcoma subtypes, underlining the particular importance of this mechanism in ULMS. Furthermore, metastatic ULMS lesions display an increased prevalence of PTEN mutations compared to primary tumors, correlating with pathway downregulation. This suggests that pathway dysregulation may contribute to disease progression [54,137].

The PI3K/Akt/mTOR pathway is less activated in ULM than in ULMS, yet it significantly contributes to ULM development [10]. Compared to normal myometrium, proteins in this pathway exhibit elevated activity in ULM. In vivo experiments demonstrate that inhibition of Akt results in a significant reduction in tumor growth [2,138]. A potential genomic cause for PI3K/Akt/mTOR pathway upregulation in ULM could be the frequent deletion of *COL4A5* and *COL4A6* genes, leading to upregulation of IRS4 (insulin receptor substrate 4) (Figure 2, Ia) [139]. High IRS4 levels in cancer cells activate the PI3K/Akt/mTOR pathway, and experimental knockdown of IRS4 significantly decreases pathway proteins, suggesting its potential regulation in ULM. This process contributes to cell proliferation, tumor growth, and resistance to various cancer therapies, including HER2-targeted therapies in breast cancer and EGFR-TKIs in non-small cell lung cancer [140,141].

#### 3.2.3. Epigenetic Regulation

HMGA1, a non-histone chromatin protein frequently overexpressed in cancers, provides an important epigenetic regulatory mechanism. High HMGA1 levels correlate with poor outcomes in trabectedin-treated ULMS patients. Experimental silencing of HMGA1 enhances trabectedin sensitivity while reducing PI3K/Akt/mTOR pathway activity (Figure 2, IIc). Combining trabectedin with mTOR inhibitors shows synergistic antitumor effects and decreases HMGA1 expression, revealing a regulatory relationship between HMGA1 and PI3K signaling, and identifying a potential therapeutic target in ULMS [142].

BRD9 overexpression also contributes to pathway activation through epigenetic mechanisms. Studies show BRD9 overexpression in ULMS compared to normal myometrium and benign ULM, making it a potential therapeutic target. BRD9 inhibition suppresses mTORC1 pathway activity, suggesting it contributes to PI3K/Akt/mTOR pathway upregulation in ULMS (Figure 2, IId) [50]. ULM also shows BRD9 overexpression compared to normal myometrium, likely driving increased PI3K/Akt/mTOR pathway activation similarly (Figure 2, Ic) [49]. This BRD9-aggressiveness-PI3K/Akt/mTOR relationship extends beyond ULMS, also observed in gallbladder cancer [143].

#### 3.2.4. MicroRNA-Mediated Regulation

MiRNA dysregulation constitutes an additional regulatory mechanism influencing PI3K/Akt/mTOR pathway activation in both ULM and ULMS.

MiR-34a serves as a valuable discriminatory biomarker between ULMS and benign ULM, with ULMS specimens consistently exhibiting substantially diminished expression levels [96]. Notably, miR-34a demonstrates significant expression variability between patients responsive and non-responsive to eribulin therapy in ULMS. In CRC, miR-34a has been shown to function as a tumor suppressor by downregulating the PI3K/Akt/mTOR signaling pathway (Figure 2, IIe) [144]. Similar regulatory mechanisms have been observed in other cancers, including gastric and lung cancer, suggesting a broader role for miR-34a in cancer progression through modulation of multiple critical signaling pathways, including PI3K/Akt/mTOR [145]. Similarly, miR-152 demonstrates decreased expression in ULMS, and experimental miR-152 overexpression in LMS cell models results in PI3K/Akt/mTOR pathway inhibition, suggesting a mechanistic link between miR-152 downregulation and pathway activation in ULMS (Figure 2, IIf) [89,95].

In ULM, several miRNAs also regulate this pathway. Studies suggest that reduced expression of miR-15 and miR-16 is associated with distinct patterns of ULM tumorigenesis, with their upregulated target genes involved in key pathways, including mTOR signaling (Figure 2, Id) [100]. The miR-200 family similarly acts as tumor suppressors by regulating multiple genes in the Akt signaling pathway, as does miR-29b (Figure 2, Ie) [62].

#### 3.2.5. Pathway Crosstalk

The interplay between the Wnt/β-catenin and PI3K/Akt/mTOR pathway in ULM is especially prominent in LSC. Wnt4, highly expressed in intermediately differentiated ULM cells, activates the PI3K/Akt signaling pathway in LSC. The function of Wnt4 appears to be Akt-dependent, with Wnt4-induced β-catenin phosphorylation being blocked by Akt inhibition, suggesting a critical regulatory relationship (Figure 2, Ib) [128,146].

#### 3.2.6. Hormonal Regulation Interface

Another layer of complexity involves the interplay between the PI3K/Akt/mTOR pathway and hormonal regulation. In ULM, estrogen signaling leads to the activation of growth-associated proteins, including PI3K [1]. Cell line studies have demonstrated that PI3K or mTOR inhibitors can suppress estrogen-induced proliferation in ULM, highlighting the pathway’s importance in hormone-mediated growth [147].

The role of progesterone signaling reveals important mechanistic insights into pathway activation. ULM tissue expresses higher levels of PR compared to normal myometrium [1,148]. Mechanistically, progesterone promotes ULM growth through rapid, non-genomic activation of the PI3K/Akt pathway. Experimental studies demonstrate that progesterone treatment increases Akt phosphorylation, an effect that is blocked by progesterone receptor antagonists, confirming receptor dependence. PI3K inhibition prevents progesterone-induced Akt activation and demonstrates the pathway’s essential role in mediating progesterone’s growth-promoting effects. Activated Akt then leads to downstream phosphorylation of GSK3β and FOXO1, with FOXO1 translocation from nucleus to cytoplasm [149]. This mechanistic understanding could explain why progesterone receptor inhibition reduces bleeding symptoms in ULM patients but does not consistently reduce tumor volume. Its primary effect is on receptor-mediated rapid signaling rather than long-term growth suppression.

Collectively, these findings demonstrate that PI3K/Akt/mTOR pathway dysregulation in uterine tumors results from complex interactions between genetic alterations, hormonal influences, and notably, emerging epigenetic mechanisms that represent novel regulatory layers, establishing it as a critical hub for therapeutic intervention.

### 3.3. TGF-β Signaling

TGF-β signaling, while less prominent in ULM and ULMS, plays a crucial role due to its complex interactions with the PI3K/Akt pathway and context-dependent effects. The pathway exhibits the classic “TGF-β paradox” with fundamentally opposing roles depending on cellular context.

In normal cells and early tumor stages, TGF-β acts as a tumor suppressor via the classical SMAD2/3 (small mothers against decapentaplegic) pathway. However, cancer cells often overcome these suppressive effects through mutations in TGF-β receptors, SMAD proteins, or enhanced PI3K/Akt signaling that blocks growth-inhibitory responses. Once this tumor suppressive function is lost, TGF-β switches to a tumor-promoting role through both SMAD-dependent and non-SMAD pathways. This dual nature makes TGF-β both a guardian and a saboteur in cancer development [150,151,152]. The molecular basis of this paradox lies in pathway crosstalk. TGF-β activates PI3K/Akt through non-SMAD pathways, alternative signaling routes that operate independently of the classical SMAD2/3 transcriptional cascade. Conversely, activated Akt promotes TGF-β receptor trafficking to the cell surface, enhancing TGF-β sensitivity. This creates positive feedback loops particularly relevant in cancer cells where the PI3K pathway is already hyperactivated, making these cells hypersensitive to TGF-β and promoting aggressive tumor behavior. The PI3K/Akt/mTOR pathway collaborates with TGF-β signaling to complete EMT by promoting protein synthesis required for cellular reprogramming and providing survival signals that prevent apoptosis during the mesenchymal transition [153,154,155,156].

The TGF-β paradox is evident in uterine tumors. In normal myometrial smooth muscle cells, TGF-β functions as a potent tumor suppressor. However, in ULM, TGF-β3 and TGF-β1 are overexpressed compared to normal myometrium and switch to a tumor-promoting role [157]. TGF-β3 is particularly prominent in driving cellular hypertrophy, increasing collagen expression, and enhancing ECM synthesis. This contributes to the characteristic ECM accumulation and tissue stiffening seen in ULM [158,159]. Multiple epigenetic mechanisms underlie this dysregulation. TET3, which is upregulated in ULM, binds to TGFBR2 (TGF-β receptor 2) promoter regions and increases TGFBR2 expression through DNA demethylation. TET3 knockdown reduces ULM cell proliferation, likely by reducing TGF-β3 receptor signaling (Figure 2, If) [26]. MiR-29c, downregulated in ULM, contributes to hormone- and TGFβ-dependent ECM accumulation, while higher miR-29 levels inhibit collagen gene expression (Figure 2, Ig) [26]. LncRNA H19, increased expressed in ULM, promotes TGF-β signaling and ECM synthesis (Figure 2, Ih) [79].

Hormonal integration reveals that progesterone upregulates TGF-β3 expression, creating an amplification cascade. Conversely, calcitriol suppresses TGF-β3 production and proliferation, providing mechanistic insight into why vitamin D deficiency represents a ULM risk factor [17,160].

While TGF-β signaling has been understudied in ULMS research compared to ULM, emerging evidence reveals its critical role in promoting malignant transformation and EMT in sarcoma biology. Experimental studies demonstrate that TGF-β1 treatment of ULMS cells activates canonical signaling and promotes EMT. TGF-β pathway blockade reverses these effects and reduces cell proliferation. The relationship between CBR1 (Carbonyl Reductase 1) and TGF-β signaling provides crucial insights into ULMS pathobiology. Experimental CBR1 overexpression in ULMS cells suppresses TGF-β production and disrupts downstream signaling in ULMS cells, effectively breaking the autocrine TGF-β loop that these cancer cells depend on for their aggressive behavior (Figure 2, IIg). The mechanism, however, remains unclear. Exogenous TGF-β treatment can still activate signaling in CBR1-overexpressing cells, suggesting that CBR1 primarily inhibits endogenous TGF-β production rather than directly disrupting receptor-mediated signaling. This demonstrates that ULMS cells are highly dependent on autocrine TGF-β signaling for maintaining their aggressive phenotype, and that CBR1 acts as a natural tumor suppressor by disrupting this autocrine loop [161].

## 4. Potential Biomarkers from Epigenetic Profiles of Uterine Leiomyomas and Leiomyosarcomas

While several markers have been established to distinguish between ULM and ULMS in tissue samples, emerging liquid biopsy technologies, particularly for miRNA detection, could revolutionize differential diagnosis of ULMS and ULM through non-invasive procedures. This approach could also establish the foundation for improved risk stratification of STUMP (smooth muscle tumors of uncertain malignant potential), which currently presents a significant diagnostic challenge due to its ambiguous biological behavior.

Distinct miRNA signatures demonstrate considerable potential for differential diagnosis. MiR-221 and miR-206 exhibit divergent expression patterns in tissue samples, providing a foundation for future serum-based analyses and diagnostic tool development. Additional miRNAs have shown differential expression in tissue samples, including miR-144-3p and miR-34a-5p, with the latter demonstrating downregulation in both tumor types but more pronounced reduction in ULMS compared to ULM. A diagnostic panel combining miR-1246 and miR-191-5p has already been validated for distinguishing ULMS from ULM in serum samples, demonstrating the feasibility of liquid biopsy approaches. Validation of tissue-based findings in serum samples could pave the way for comprehensive liquid biopsy platforms. As liquid biopsy technologies continue to advance, methodologies are becoming increasingly cost-effective and efficient. Identification of individual miRNAs or miRNA combinations that achieve high sensitivity and specificity for differential diagnosis could substantially transform ULM and ULMS diagnostics while potentially avoiding unnecessary invasive procedures.

Detection of DNA methylation patterns in liquid biopsies represents another promising avenue for ULMS and ULM diagnosis [162]. However, as previously discussed, this approach requires prior completion of prospective, multi-center comparative DNA methylation studies with adequate sample sizes for both ULM and ULMS populations. The integration of serum miRNA signatures and DNA methylation profiles with radiological findings of uterine smooth muscle tumors shows considerable promise for enhancing pre-surgical risk stratification, potentially addressing both delayed ULMS diagnosis and unnecessarily radical interventions for benign ULM.

## 5. Potential Therapeutic Implications

Epigenetic dysregulation in ULMS and ULM offers promising new therapeutic opportunities. Although most studies remain in preclinical or early-phase settings, several classes of epigenetic drugs have shown activity (Table 3). EZH2 inhibitors, both as monotherapy and in combination with a PI3K/mTOR inhibitor, have shown promising results in vitro and in vivo in ULM and ULMS, with EZH2 overexpression observed in ULM [163,164,165]. Tazemetostat, the only FDA-approved EZH2 inhibitor, is currently indicated for epithelioid sarcoma but has not yet been tested in ULM or ULMS [166,167]. DNA methyltransferase inhibitors have shown efficacy in both ULM and ULMS models in vitro and in vivo [3,27,33,163]. HDAC inhibitors such as Mocetinostat have been evaluated clinically with a Phase II trial combining Mocetinostat with Gemcitabine in metastatic LMS showed modest clinical activity [49,50]. Bromodomain and extraterminal (BET) inhibitors have shown promise in early clinical trials for various cancers but are limited by a narrow therapeutic window [164,168]. Combination strategies with BET inhibitors have shown promising results in ULMS cell lines [162,165,169,170,171]. Recent cancer research focuses on restoring tumor-suppressive miRNAs while inhibiting oncogenic ones. This therapeutic approach shows potential for ULM and ULMS treatment, supported by the established pathogenic role of miRNAs and preclinical evidence of specific miRNA activity (miR-148a-3p, miRNA-199a-5p) in ULM [172,173,174,175].

The marked difference in ALT activity between ULM and ULMS may serve not only as a diagnostic marker to distinguish benign from malignant tumors, but also as a prognostic factor in ULMS and a potential therapeutic vulnerability, as ATRX-deficient cells exhibit heightened sensitivity to replication stress and DNA damage—features that can be targeted with agents such as PARP inhibitors, WEE1 inhibitors, and G-quadruplex stabilizers; however, this has not yet been achieved in ULM or ULMS in vivo or in vitro experiments [176,177,178].

**Table 3 cancers-17-02610-t003:** Epigenome-altering drugs tested in ULMS or ULM.

Tumor	Drug	Level	Sample Type	Results	References
DNA Methyltransferase Inhibitors					
ULMS	5-Azacitidine	in vitro	Established cell lines (SK-UT1, SK-LMS1, MES-SA)	- Reduced Cell Viability - Anti-proliferative Effects: Increased Apoptosis, Cell Cycle Arrest	[163]
	5′-Aza-2′-Deoxycytidine	in vitro		- Reduced Cell Viability - Anti-proliferative Effects - Inhibition of DNMT1 protein expression	
		in vitro		- Reduced proliferation and migration - Induced cell cycle arrest and apoptosis - Downregulated DNMTs, Hedgehog pathway, and proliferation markers	[27]
	Guadecitabine	in vitro/in vivo	Established cell lines (SK-UT1, SK-LMS1, MES-SA) Xenograft model (SK-UT1, SK-LMS1 in NOD/SCID mice)	- Superior cytotoxicity compared to Aza/DAC - Inhibited growth and induced apoptosis - Extended survival duration	[163]
ULM	5′-Azacitidin	in vitro/in vivo	Primary ULM cells, Xenograft model (Primary human leiomyoma cells in NSG mice)	- Induced LSC differentiation and reduced stemness - Reprogrammed transcriptome toward differentiation - Showed synergistic tumor suppression with antiprogestin	[31,33]
	5′-Aza-2′-Deoxycytidine	in vitro	Primary ULM cells	- Reduced proliferation without apoptosis - Inhibited ECM formation and Wnt signaling - Upregulated tumor suppressor genes (*KLF11*, *DLEC1*, *KRT19*)	[179,180]
HDAC Inhibitors					
ULMS	Tucidinostat	in vitro	Established cell lines (SK-UT-1 and MES-SA)	- Inhibited proliferation and induced apoptosis/cell cycle arrest - Modulated multiple signaling pathways and gene expression - Altered histone modifications) - Increased EMT levels	[181]
	DL-sulforaphane	in vitro	Established cell lines (SK-UT-1 and MES-SA)	- Inhibited proliferation and induced apoptosis - Downregulated cell cycle regulators and HDAC6 - Modulated signaling pathways and histone modifications - Increased EMT levels	
	Mocetinostat (in combination with gemcitabine)	Phase II	Clinical trial, patients with metastatic LMS (including 8/20 with ULMS)	- Common toxicities: fatigue, cytopenia - No reversal of gemcitabine resistance - Modest activity: 1 Partial Response (in a patient with ULMS), 12 Stable Diseases (5 of them in ULMS), 5 Progressive Diseases, short Progressive Free Survival (2.0 months) - Study discontinued due to limited response rates	[172]
ULM	Vorinostat	in vitro	Primary ULM cells	- Significant reduction in cell viability	[173]
EZH2 inhibitors					
ULM	3-Deazaneplanocin A (DZNep)	in vitro	Primary ULM cells	- Upregulated DNA damage repair genes (*RAD51, BRCA1*) - Substantially inhibited proliferation - Caused G2/M arrest without apoptosis	[44]
Bromodomain Inhibitors					
ULMS	BET inhibitors (JQ1, I-BET 762, GS-626510)	in vitro/in vivo	Established cell line (SK-UT-1), Xenograft model (Patient-derived ULMS tumors in SCID mice)	- Inhibited viability and induced cell cycle arrest - Altered transcriptome and reduced metastasis-related transcription factors - Modified epigenetic regulators and critical signaling pathways	[171,174]
ULM	TP-472 (BRD9 inhibitor)	in vitro	Immortalized ULM and uterine smooth muscle cell lines	- Increased apoptosis, arrested proliferation, reduced ECM - Affected multiple pathways: cell cycle, inflammation, E2F targets, ECM deposition, m6A reprogramming	[49]
	miRNAs therapies				
ULM	Anti-miR-148a-3p	in vitro/in vivo	Primary ULM cells, Xenograft model (Patient-derived ULM tumor in NOD SCID mice)	- Inhibition of proliferation of patient-derived leiomyoma cells and tumor growth in vivo	[175]
	miRNA-199a-5p	in vitro	Primary ULM cells	- OE of miRNA-199a-5p inhibited primary ULM proliferation	[182]

Targeting the biological pathways modulated by epigenetic alterations is also a potential strategy in ULM and ULMS (Table 4). Inhibition of the PI3K/AKT/mTOR pathway has demonstrated antitumor activity, with Temsirolimus, an mTOR inhibitor, inducing a partial response in one ULMS case [183,184,185]. BEZ235, a dual PI3K/mTOR inhibitor, in combination with Doxorubicin, exhibited synergistic effects in vitro and in vivo for ULMS [186]. Wnt/β-catenin signaling pathway inhibitors have shown antiproliferative effects in ULM in vitro studies [180,187,188,189]. Limited studies have demonstrated efficacy of TGF-β signaling inhibition in ULM, with the most recent published in 2022 [190]. However, a major challenge in targeting pathways such as Wnt/β-catenin is their essential role in development, tissue maintenance, and homeostasis, which significantly limits the therapeutic window. Additionally, extensive crosstalk with non-canonical and other signaling pathways—such as the Notch, Hippo, and TGF-β pathways—can promote compensatory mechanisms and contribute to drug resistance (e.g., Wnt/β-catenin and Notch crosstalk) [191].

**Table 4 cancers-17-02610-t004:** Pathway inhibitors.

Tumor	Drug	Level	Sample Type	Results	References
Inhibition of the PI3K/mTOR pathway					
ULMS	BEZ235 (Dual PI3K/mTOR inhibitor)	in vitro	Primary ULMS cells	- Synergistic growth inhibition and apoptosis induction, more effective than each inhibitor individually in combination	[192]
		in vitro/in vivo	Established cell lines (SK-LMS-1, STS39), xenograft model (SK-LMS-1 in NSG mice)	- Inhibition of downstream signaling pathway effector - Combination with doxorubicin: Synergistic growth inhibition, induction of apoptosis - Tumor growth inhibition by BEZ235 alone and enhanced by the combination with doxorubicin	[186]
	Temsirolimus (mTOR inhibitor)	Phase I and Phase II	Patients with advanced STS	- Phase I: 4 LMS, no objective response seen - Phase II: 9 LMS, with 1 partial response seen with a ULMS patient	[184,185]
ULM	MK-2206 (Akt-inhibitor)	in vitro/in vivo	Primary ULM and myometrial cells, xenograft model (primary ULM cells in immunocompromised mice)	- Reduced the cell viability, cell death independent of caspase activity - Significantly reduced the tumor volume	[193]
Inhibition of the Wnt/β-catenin signaling pathway					
ULM	ICAT, Niclosamide, XAV939	in vitro	Primary ULM cells	- Inhibits proliferation, without affecting cell viability - Inhibits Wnt/β-catenin signaling pathway - Significantly nuclear translocation of β-catenin	[194]
Inhibition of the TGF β signaling pathway					
ULM	LY364947 (TGF-β1 Inhibitor)	in vitro	Primary ULM and myometrium cells	- Significant reduction in cell viability to 27% (vs. 49% in normal myometrium) - More specific effect on leiomyomas than normal tissue	[190]

## 6. Future Directions

Women’s health has historically been underrepresented in biomedical research, largely due to concerns over hormonal confounding factors [195]. This exclusion has led to critical diagnostic and therapeutic gaps, particularly in conditions like uterine smooth muscle tumors. Symptoms related to the uterus are still often dismissed as normal menstrual variation rather than recognized as signs of pathology. This literature review addresses that gap by examining epigenetic mechanisms as a lens for improving diagnostic and therapeutic precision in uterine leiomyoma (ULM) and leiomyosarcoma (ULMS).

Our comparative analysis of DNA methylation, histone modifications, and non-coding RNAs in ULM and ULMS revealed surprisingly convergent downstream effects, despite distinct epigenetic alterations. Notably, both tumor types impact key oncogenic pathways. PI3K/AKT/mTOR, Wnt/β-catenin, and TGF-β signaling, highlighting a novel insight: diverse genetic and epigenetic mechanisms, operating at different regulatory levels, can ultimately activate the same oncogenic signaling pathways and lead to functionally equivalent outcomes.

MicroRNAs (miRNAs) represent a dual opportunity as non-invasive biomarkers for diagnosis and as therapeutic targets. Future research should explore miRNA-based liquid biopsy panels, especially those combining ULMS-specific and ULM-distinct miRNAs, such as miR-221, miR-206, and miR-1246. A potential study design could include a prospective validation of these panels in serum and tissue across ULM, STUMP, and ULMS cases.

Similarly, DNA methylation profiling holds promise not only for understanding tumor biology but also for clinical application. However, large-scale, prospective studies are needed. A proposed multicenter study should include at least 50 samples per group (ULM and ULMS), utilizing standardized methylation arrays and correlating epigenetic signatures with histopathological and clinical data, including response to therapy.

Given the documented DNMT1 overexpression and PI3K/mTOR pathway activation in ULMS, a combination approach targeting both axes may offer therapeutic synergy. We therefore suggest that a phase I clinical trial investigating demethylating agents in combination with mTOR inhibitors—particularly in DNMT1-high ULMS patients—could represent a promising direction for translational research.

BRD9 overexpression in both ULM and ULMS further supports the concept of a progressive epigenetic continuum between the two tumor types. While BRD9 inhibition induces G1 arrest and suppresses proliferation in ULM, ULMS cells respond with broader transcriptional shifts and apoptosis. This shared vulnerability suggests BRD9 may serve as both a therapeutic target and a potential marker for malignant transformation from ULM to ULMS. However, while BET inhibitors have shown promising preclinical activity in ULMS models, their role in ULM remains largely unexplored. Systematic investigation of BET inhibition in ULM is urgently needed to determine whether similar epigenetic dependencies exist and could be exploited therapeutically.

In addition, the frequent loss or mutation of ATRX in ULMS, but not in ULM, offers another therapeutic opportunity. ATRX deficiency leads to alternative lengthening of telomeres (ALT), rendering tumors particularly vulnerable to replication stress. This makes ATRX-mutant ULMS candidates for synthetic lethality-based approaches, such as treatment with PARP inhibitors, WEE1 inhibitors, or G-quadruplex stabilizers. Although preclinical evidence from other cancers supports this rationale, targeted studies in ULMS are still lacking and should be prioritized.

Whether ULMS arises de novo or evolves from ULM remains unresolved. Epigenetic analyses reveal marked differences between the two tumor types, yet pathway activity gradients (e.g., increasing PI3K/AKT activation from ULM to STUMP to ULMS) suggest a possible continuum. MiRNA profiles, particularly the progressive downregulation of tumor-suppressive families like let-7 and miR-200, support this model. Further studies using methylome sequencing in morphologically ambiguous tumors may clarify this transition hypothesis.

In summary, our findings highlight the complexity of epigenetic regulation in uterine smooth muscle tumors and its clinical potential. ULM and ULMS differ epigenetically, yet converge on shared signaling outputs, offering a unique opportunity for biomarker development and therapeutic repurposing. Closing the research gap in women’s health, especially through the lens of the epigenome, is not only scientifically urgent but also a matter of healthcare equity.

## 7. Conclusions

This literature review has synthesized current knowledge of epigenetic aberrations in ULM and ULMS, contextualizing these findings to identify interconnected mechanisms and shared therapeutic vulnerabilities across different epigenetic landscapes. By uncovering their convergent activation of identical pathways, this analysis provides rationale for future investigations combining hormonal interventions with epigenetic therapies and pathway inhibitors in comprehensive treatment strategies. Future progress requires increased funding for larger cohort studies, especially in rare ULMS, to validate these epigenetic findings and enable clinical translation.

## Figures and Tables

**Figure 1 cancers-17-02610-f001:**
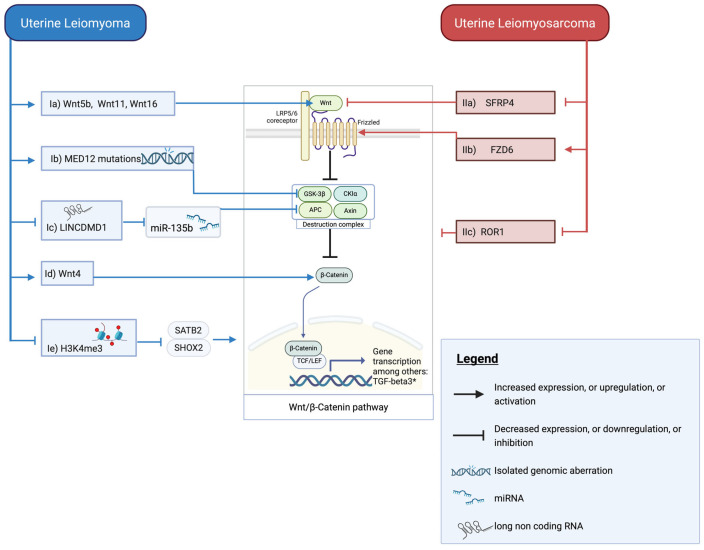
Interaction with the Wnt/β-catenin pathway: ULM(I) and ULMS(II) epigenomic and genomic alterations for upregulation of the Wnt/β-catenin pathway. (Ia) Increased expression of Wnt ligands leads to enhanced pathway activation. (Ib) *MED12* mutations are associated with upregulation of the Wnt pathway. (Ic) Decrease in LINCMD1 leads to an increase in miR-135b, inhibiting *APC* and therefore weakening the destruction complex. (Id) Increased Wnt4 is associated with an increase in β-catenin. (Ie) Global reduction in the histone mark H3K4me3 leads to increased expression of the pathway activators *SATB2* and *SHOX2*. (IIa) Low SFRP4 leads to less antagonization of Wnt ligands, therefore upregulating the pathway. (IIb) An increase in members of the Frizzled receptor family leads to more downstream pathway activation. (IIc) ROR1, part of the non-canonical Wnt/β-catenin pathway, which normally downregulates the canonical Wnt/β-catenin, shows decreased expression in ULMS.

**Figure 2 cancers-17-02610-f002:**
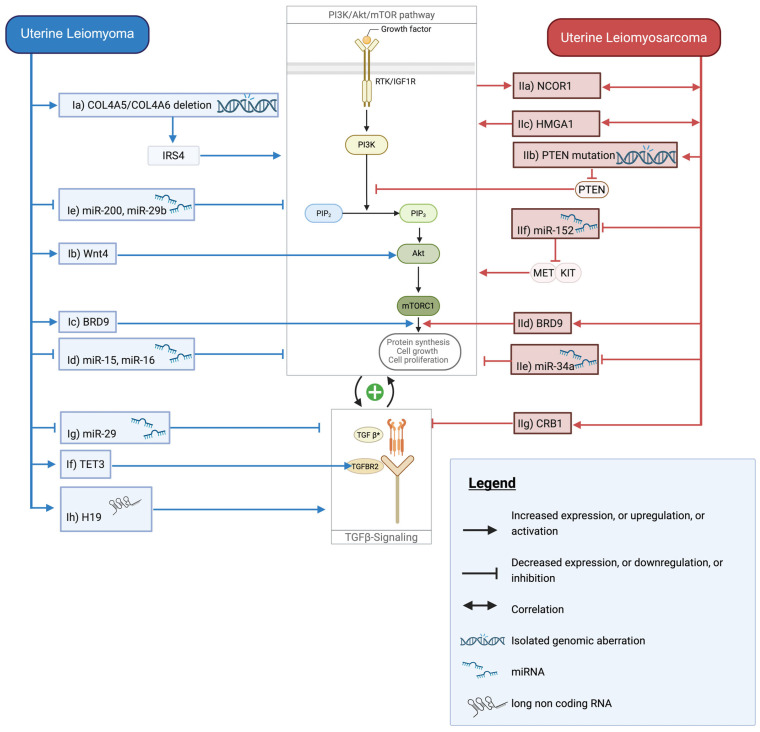
Interaction between the PI3K/Akt/mTOR pathway and TGF-β-signaling: Regulation of the PI3K/Akt/mTOR pathway and TGF-β signaling by aberrations in ULM (I) and ULMS (II). ULM: (Ia) Genomic *COL4A5/COL4A6* deletion leads to increased expression of IRS4, which upregulates PI3K pathway. (Ib) Wnt4 leads to an increase in active Akt in LSC. (Ic) BRD9 overexpression increases mTOR pathway activity. (Id) Decreased miR-15/-16 results in reduced inhibition of the PI3K pathway. (Ie) Decreased miR-200 and miR-29b result in reduced inhibition of the PI3K pathway. (If) TET3 demethylates the promotor region of TGFBR2, leading to increased expression. (Ig) Reduced miR-29 decreases inhibition of TGF-β signaling. (Ih) H19 is increased and promotes TGF-β signaling. ULMS: (IIa) NCOR1, characteristic of more aggressive ULMS, is upregulated by the PI3K/Akt/mTOR pathway. (IIb) Mutations in *PTEN* lead to decreased expression of the PI3K inhibitor PTEN. (IIc) HMGA1 is associated with the PI3K pathway. (IId) BRD9 overexpression increases mTOR pathway activity, similar to ULM. (IIe) Decreased miR-34a results in reduced inhibition of the PI3K pathway (as seen in CRC). (IIf) Decreased miR-152 leads to upregulation of MET and KIT, both activators of the PI3K pathway. (IIg) Association between CBR1 and TGF-β signaling.

**Table 1 cancers-17-02610-t001:** Specific gene methylation pattern between ULMS and ULM—all mentioned studies investigated tissue samples.

Gene	Methylation Status			Function	References
	ULMS	ULM	Normal Myometrium		
NPAS4	hypermethylated	unmethylated	unmethylated	Transcription factor	[29]
PITX1	unmethylated	hypermethylated	unmethylated	TSG (Tumor suppressor gene)	
*KLF4*	hypermethylated	hypermethylated *	unknown	TSG (mainly)	[15,36]
*DLEC1*	hypermethylated	hypermethylated *	unknown	TSG	
*MGMT*	no difference	no difference	no difference	TSG	[36]
*BCL2*				Oncogene	
*EGR1*				Unclear	
*TIMP3*				Potential TSG	
*BIRC5*				Potential oncogene	
*ANXA11*				Unclear	
*CTGF*				Unclear	
*COL4A1*	unknown	hypomethylated	unknown	ECM-Genes (extracellular matrix genes)	[37,38]
*COL4A2*					
*COL6A3*					
*FAM9A*				X-Chromosome-Gene, function unclear	
*CPXCR1*					
*CXORF45*					
*TAF1*					
*NXF5*					
*KRT19*				TSG	[12]
*KLF11*					
*KAT6A*	hypermethylated	unknown	unknown	Chromatin-modifying genes	[34]
*KMT2A*					
*EZH2*					
*CTNNB1*				Chromatin/DNA-binding-genes	
*PBX3*					
*SATB1*					
*MEIS*					
*COMMD1-BMI*					

* significantly more than ULMS.

**Table 2 cancers-17-02610-t002:** miRNA expression profiles in ULM and ULMS compared to normal myometrium unless otherwise specified.

miRNA	Expression Changes		Function/Target	Sample Type	References
	ULMS	ULM			
Expression partly unknown					
miR-29b miR-29c	unknown	↓	- XIST lncRNA target (miR-29c) - Estrogen/progesterone-responsive - TGF-β3 negative regulation - ECM homeostasis (miR-29b)	Tissue ULM xenograft model	[65,66,67,71,79,80]
miR-93	unknown	↓	- IL-8 inverse correlation - PTEN targeting, PI3K/Akt activation	Tissue	[18,70,71,72]
miR-27b	unknown	↓	- CYP1B1 inverse regulation, estrogen metabolism	Tissue	[66,67]
miR-152	↓	unknown	- PI3K/Akt/mTOR pathway via MET/KIT targeting	Tissue	[93,94,95]
Established similar expression changes					
let-7	↓	↓	- HMGA2 repression, proliferation inhibition - Tumor suppressor, prognostic marker	Tissue	[18,67,99,100,101,102]
miR-200	↓↓	↓	- EMT regulation - Proliferation inhibition - Apoptosis induction - ECM remodeling	Tissue established cell lines (SK-LMS-1)	[18,69,89,103,104,105]
miR-1	↓	↓	- Tumor suppressor activity	Established cell lines (SK-UT-1, THESCs, PCS-460-011)	[93,106,107,112]
miR-34a (-5p)	↓↓	↓	- Differential downregulation (ULMS > ULM) - Diagnostic biomarker potential - Therapeutic target candidate	Tissue	[96,111]
Established divergent expression changes					
miR-221	↑	↓	- Proliferation, invasion, migration promotion	Tissue	[89,90]
miR-206	↑	↓	- ERα post-transcriptional regulation - Diagnostic biomarker potential	Tissue	[70,113,114,115]
miR-1246	↓	↑	- Oncogenic function - Diagnostic biomarker potential	Serum	[109,116]
miR-191 (-5p)	↓	unknown	- Oncogenic function - Differential expression (ULMS↓ vs. ULM) - Diagnostic biomarker potential	Serum	[109]
miR-144 (-3p)	↑	unknown	- Differential expression (ULMS↑ vs. ULM) - Diagnostic biomarker potential	Tissue	[111]

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
