# Peer review of "Decoding the Epigenome: Comparative Analysis of Uterine Leiomyosarcoma and Leiomyoma"

_cancers, 2025, doi:10.3390/cancers17162610_

Round 1
Reviewer 1 Report
Comments and Suggestions for Authors
The work “Decoding the Epigenome: Comparative Analysis of Uterine Leiomyosarcoma and Leiomyoma” is devoted to the current problem of studying these diseases, since it opens up opportunities for more precise diagnostic criteria and potential targets for therapy. The review presents a large number of studies of epigenetic regulation in ULMS and ULM. The Reviewer notes the accuracy of the conclusions that the Authors draw based on the reviewed works with small amount of samples. Despite the above mentioned strengths of the work, while reading, the Reviewer had some comments and suggestions, as follows:
MAJOR
- The Reviewer recommends that the Authors add a table summarizing the miRNAs results in ULM and ULMS as well as their associations with driver mutations (MED12, HMGA2 for ULM etc.) and steroid hormones (estrogen and progesterone) as was done for gene methylation pattern.
- The description of long non-coding RNAs is the part of "ULM-specific miRNA patterns" section, which is apparently incorrect. In addition, there are many more lncRNAs associated with ULM, so the Authors should revise it in more details.
- There is no section on lncRNAs in ULMS. Based on the previous point, this information must be added.
- It is strongly recommended to include information in chapter 5 on the potential therapeutic implications of epigenetic therapy for ULM using non-coding RNA delivery. Several articles published since 2022 should be considered.
MINOR
The Authors should review their manuscript for misspells and errors (e.g. line 304 contains an unnecessary ":")
Author Response
Response to Reviewer 1:
We appreciate the reviewer's valuable comments and suggestions, which have helped improve the quality and clarity of our manuscript. Please find our point-by-point responses below.
- The Reviewer recommends that the Authors add a table summarizing the miRNAs results in ULM and ULMS as well as their associations with driver mutations (MED12, HMGA2 for ULM etc.) and steroid hormones (estrogen and progesterone) as was done for gene methylation pattern.
As recommended, we have added a table summarizing the most important miRNAs in ULM and ULMS with their associations with the hormone axis and driver mutations. We combined associations with hormones and driver mutations into a single column, as information was available for only a few miRNAs and we wanted to maintain a clear and concise presentation
- The description of long non-coding RNAs is the part of "ULM-specific miRNA patterns" section, which is apparently incorrect. In addition, there are many more lncRNAs associated with ULM, so the Authors should revise it in more details.
The reviewer was absolutely correct that placing lncRNAs in the miRNA section was confusing for readers. We originally structured it this way to maintain reading flow and immediately connect interactions such as miR-29 and XIST. Following the reviewer's recommendation, we have reorganized the content and established appropriate subheadings for better clarity. Our initial selection of lncRNAs was based on multiple mentions across different publications to ensure credibility. We have expanded this section to include additional lncRNAs that have been identified as important for ULM pathogenesis.
- There is no section on lncRNAs in ULMS. Based on the previous point, this information must be added.
As requested, we have added a dedicated section on lncRNAs in ULMS. While research in this area remains limited, we have included the available ULMS-specific data, including EGFR-AS1 and relevant prognostic signatures, while acknowledging the need for more targeted studies in this field.
- It is strongly recommended to include information in chapter 5 on the potential therapeutic implications of epigenetic therapy for ULM using non-coding RNA delivery. Several articles published since 2022 should be considered.
Thank you for this valuable suggestion. Since 2022, several articles have characterized non-coding RNAs in ULM with potential therapeutic implications, however, only a few studies that we identified conducted direct therapeutic experiments. To improve the flow of the article, we included those with a direct therapeutic angle. We added this information to the text and our table. One of these, miR-148a-3p, particularly has robust in vivo and in vitro data. We did however not include those miRNAs in the miRNA chapter because they have each been discussed in only 1-2 papers, and we focused on the most prominent ones that have been replicated across multiple studies.
- The Authors should review their manuscript for misspells and errors (e.g. line 304 contains an unnecessary ":")
We thank the reviewer for pointing out these issues. We have thoroughly proofread the manuscript and corrected all spelling errors and typographical mistakes, including the unnecessary colon on line 304.
We believe these revisions have significantly strengthened the manuscript and addressed all the reviewer's concerns comprehensively. We are grateful for the opportunity to improve our work and look forward to your further evaluation.

Reviewer 2 Report
Comments and Suggestions for Authors
The introduction provides a solid clinical and molecular context for differentiating between uterine leiomyomas (ULM), leiomyosarcomas (ULMS), and STUMP. However, the structure and flow would benefit from clearer organization. In particular, the discussion on the potential progression from ULM to ULMS is repeated and somewhat disjointed. This could be streamlined by fully addressing the genomic controversy before transitioning to epigenetic mechanisms.
There’s also a sentence construction issue: “While genomic analyses have not definitively resolved this Emerging evidence has shown...” is grammatically incorrect and interrupts the flow. It should read: “While genomic analyses have not definitively resolved this, emerging evidence has shown...”
Additionally, the transitions between clinical background, molecular mechanisms, and the aims of the review are abrupt. Introducing brief linking sentences would improve readability. For example, before discussing epigenetics, a sentence like “This uncertainty has led researchers to explore epigenetic mechanisms as alternative diagnostic tools” would clarify the rationale.
Some sentences are overly long and could be split for clarity, especially in the paragraph on ULMS prognosis and treatment response. Finally, the objective of the review is implied but not explicitly stated concisely—adding a clear closing sentence outlining the purpose (e.g., identifying epigenetic biomarkers and therapeutic targets) would strengthen the conclusion of the introduction.
I would recommend citing the most recent ESGO guidelines in the introduction, and placing greater emphasis on the preoperative diagnostic challenges associated with ULMS (e.g., PMID: 38054268; doi:10.1002/uog.20270)."
- Epigenetic alterations
2.1. DNA Methylation Landscapes
This section offers a thorough summary of current findings on DNA methylation in uterine smooth muscle tumors and highlights its potential as a diagnostic and therapeutic tool. However, there are several areas that require improvement in clarity, structure, and interpretation.
First, the narrative suffers from redundancy and lack of synthesis. Numerous studies are reported consecutively, often without critical integration or highlighting of overarching trends. For example, both Liu et al. and Miyata et al. are cited to support global hypomethylation in ULMS, but their conclusions are not clearly reconciled. The manuscript should focus less on listing study details and more on synthesizing their collective implications.
Sample size limitations are correctly acknowledged, but the text tends to overstate conclusions from underpowered studies. For instance, the statement that ULMS shows consistent CpG island hypermethylation is weakened by conflicting data (e.g., from Conconi et al.). The review should emphasize that current evidence is preliminary and sometimes contradictory, and should more strongly advocate for larger, well-controlled methylation studies.
Language and sentence structure need tightening. Several sentences are overly long or packed with technical details that obscure the main point. For example:
“This hypomethylation was particularly prominent in intergenic regions, gene bodies, 3'-UTR regions, and TSS1500 regions but less pronounced in TSS200 regions and 5'-UTR/first exon categories.”
This could be rewritten more clearly as:
“Hypomethylation in ULMS predominantly affects intergenic regions, gene bodies, and distal promoter regions, while proximal promoters and first exons are less affected.”
The section on hormone-related methylation changes is scientifically relevant but somewhat fragmented. The link between DNA methylation and hormone receptor signaling (PGR, ERα) is well-made, but the molecular mechanisms are described in too much detail without tying back clearly to clinical significance. It would help to summarize key messages—for example, that aberrant methylation in hormone-regulated pathways may explain differences in hormone responsiveness between ULM and ULMS.
The final paragraph appropriately calls for larger studies, but it would be stronger if framed as a future research direction rather than just a knowledge gap. Instead of simply noting that ULMS is often studied as part of broader sarcoma cohorts, the authors could recommend design principles for future methylation studies that focus specifically on ULMS vs ULM.
Moderate revision is recommended to enhance clarity, reduce redundancy, and sharpen the analytical perspective.
2.2 Histone Modifications and Chromatin Remodeling Mechanisms,
This section addresses a wide range of epigenetic mechanisms beyond DNA methylation, providing a valuable overview of histone modifications and chromatin remodeling in uterine smooth muscle tumors. While the content is informative and highlights emerging targets and mechanisms, several areas require significant refinement to enhance clarity, synthesis, and scientific rigor.
First, the structure is overly dense and encyclopedic, with a succession of findings presented in rapid sequence and little attempt at integration or hierarchy. Important mechanisms—such as H3K27me3-mediated repression by EZH2, BRD9 involvement in chromatin remodeling, and ATRX loss triggering ALT—are discussed with limited connective analysis or commentary on their relative importance or potential interactions. The text would benefit from clearer transitions and grouping of findings into thematic subunits (e.g., “repressive modifications,” “hormone-epigenetic crosstalk,” “chromatin remodeler mutations”).
Redundancy and excessive detail also limit readability. Some sentences provide mechanistic minutiae that obscure the main point. For example, mentioning the precise phosphorylation mechanism of EZH2 via PI3K/AKT in the context of ER signaling may be more appropriate for a specialized mechanistic discussion than a broad tumor biology section. Similarly, the detailed breakdown of H2A.Z alterations in the rare SRCAP-mutated leiomyoma subtype could be abbreviated or moved to a footnote/reference, given its limited prevalence (~1%).
The discussion of ULM vs ULMS differences is crucial but not sufficiently emphasized or critically interpreted. While ATRX loss and ALT activation in ULMS are well described, the implications of these changes for tumor behavior and prognosis deserve stronger synthesis. For example, a comparative sentence summarizing how histone and chromatin remodeling alterations contribute to the benign vs malignant phenotypes would help contextualize the described findings.
The role of therapeutic targets (e.g., HDAC inhibitors, BRD9 inhibitors) is promising, but the therapeutic relevance is dispersed across the section without a conclusive synthesis. It would be helpful to highlight these mechanisms in a final paragraph that frames them as candidate targets, distinguishing those with preclinical validation from more speculative ones.
Finally, while the language is generally precise, sentence structure can be overly long and passive, reducing clarity. Rewriting complex ideas into shorter, active sentences would improve accessibility for readers less familiar with epigenetic terminology.
2.3.1. ULM-specific miRNA patterns 249
This section addresses an emerging and relevant topic in USMT biology. The role of miRNAs and lncRNAs is well articulated, but the presentation is fragmented and could benefit from thematic organization. Rather than listing individual miRNAs and lncRNAs, grouping them by functional axis—e.g., ECM regulation (miR-29, H19), hormonal signaling (miR-206, miR-27b, XIST), or Wnt/β-catenin pathway (LINCMD1)—would enhance clarity and biological interpretation.
The discussion on miR-29 and XIST is particularly strong, though somewhat repetitive. The interplay between hormones, miRNA expression, and downstream signaling is compelling but would be clearer with more concise language and a brief integrative summary.
The section lacks a direct comparison between ULM and ULMS non-coding RNA profiles, which would help contextualize miRNA roles in benign vs malignant progression. Mention of diagnostic potential (e.g., liquid biopsy) is promising but underdeveloped.
In short, the content is scientifically relevant and well referenced but requires structural refinement to improve readability and impact. A concluding synthesis would help frame the translational implications more effectively.
3.2 ULMS-specific miRNA patterns:
This subsection effectively highlights key miRNAs distinguishing ULMS from ULM and normal myometrium, reinforcing their potential as biomarkers and therapeutic targets. The focus on miR-221, miR-1, and miR-152 is well justified, linking their dysregulation to malignant traits and pathways like PI3K/Akt/mTOR. The cross-reference to other sarcomas strengthens the biological plausibility of their roles.
However, the mechanistic insights remain mostly inferred from other cancers, reflecting a gap in direct ULMS-specific functional studies. This limits the strength of conclusions about causality and therapeutic implications. The mention of circulating miRNAs and their impact on treatment response is promising but brief; expanding on clinical validation status would improve translational relevance.
3.1 Wnt/β-Catenin Pathway
Certain aspects could be further refined for clarity and depth. For instance, the description of epigenetic changes, such as the reduction in H3K4me3 and its downstream effects on SATB2 and SHOX2, is somewhat brief. Expanding on how these chromatin modifications mechanistically promote Wnt signaling would strengthen the reader’s understanding of the pathway’s regulation. Likewise, the role of non-canonical Wnt signaling receptors like ROR1 in ULMS is introduced but not fully developed. Providing additional context or evidence on how reduced ROR1 expression might shift signaling towards canonical β-catenin activation would enrich the discussion.
The therapeutic implications section is a valuable inclusion, effectively tying the mechanistic insights to potential clinical strategies. Nevertheless, it would benefit from a more detailed consideration of challenges involved in targeting these pathways, such as the complexity of pathway crosstalk or potential resistance mechanisms, to provide a more balanced perspective.
3.2 PI3K/AKT/mTOR Pathway 460
The section contains some lengthy and complex sentences that might hinder reader comprehension. Breaking these into shorter, clearer sentences and perhaps using subheadings or bullet points to separate mechanisms in ULM and ULMS could improve readability. Additionally, referring to Figure 1 multiple times without summarizing its key elements assumes prior familiarity and could leave some readers behind; a brief description in the text would be helpful.
Minor technical points include ensuring consistent abbreviation definitions (e.g., spelling out “gain-of-function” at first mention of MED12 mutation) and cleaning up residual editorial notations present in the text.
The section on the PI3K/AKT/mTOR pathway provides a comprehensive and well-rounded overview of the pathway's involvement in uterine tumors, specifically highlighting its progressive activation from benign ULM through intermediate STUMP to malignant ULMS. The authors effectively establish a clear correlation between increased pathway activity and tumor aggressiveness, supported by immunohistochemical and molecular data.
The detailed description of pathway activation mechanisms, beginning with ligand binding to RTKs and subsequent downstream signaling, is accurate and consistent with current understanding. The distinction made between ULMS and ULM in terms of genomic and epigenomic alterations adds depth and reflects the complexity of tumor biology. Particularly noteworthy is the discussion of PTEN mutations and NCOR1 amplification in ULMS, which underlines important drivers of PI3K/AKT/mTOR dysregulation. The mention of IRS4 upregulation as a potential genomic driver in ULM is also insightful.
Furthermore, the integration of epigenetic regulators, such as the role of NCOR1-dependent HDAC3 activity, and the involvement of microRNAs in modulating pathway activity, adds a valuable layer of mechanistic understanding. The data connecting microRNA dysregulation (notably miR-34a, miR-152, miR-15/16, and miR-200 family) to pathway activation and tumor behavior are compelling, though a more cohesive narrative linking these various miRNAs would improve clarity.
The interplay between the PI3K/AKT/mTOR pathway and hormonal signaling pathways, particularly the role of estrogen and progesterone, is well highlighted and provides an important clinical context, given the hormone responsiveness of uterine tumors. However, the complex and sometimes contradictory effects of progesterone receptor signaling on tumor growth could be discussed in more detail to clarify this important aspect.
While the section is rich in detail and well referenced, the text could benefit from improved structural organization. Currently, many concepts are presented densely within single paragraphs, which might challenge readers less familiar with the topic. Dividing the content into clearly delineated subsections or thematic paragraphs (e.g., genomic alterations, epigenetic regulation, microRNA involvement, hormonal regulation) would enhance readability and comprehension
TGF-β signaling
The section offers a thorough overview of TGF-β signaling in uterine tumors, highlighting its interactions with the PI3K/Akt/mTOR pathway and its dual role in tumor biology. However, the text could benefit from a clearer structure to improve readability and flow. Currently, the information is densely packed, which makes it challenging to follow the logical progression of ideas, especially for readers who may not be experts in this signaling pathway. Organizing the content into more distinct thematic paragraphs—such as an introduction to the pathway, epigenetic regulation, hormonal influences, and therapeutic implications—would enhance clarity and engagement.
Furthermore, while the section recognizes that TGF-β is less studied in ULMS compared to ULM, the discussion around ULMS remains brief and underdeveloped. The example of CBR1's role in modulating TGF-β signaling in ULMS is interesting but deserves a deeper exploration to provide a fuller understanding of how TGF-β contributes to ULMS pathobiology. This imbalance between ULM and ULMS content limits the comprehensiveness of the review.
The explanation of mechanistic interactions, particularly the crosstalk between TGF-β and PI3K/Akt pathways, is somewhat superficial. Although Akt phosphorylation of SMAD proteins is mentioned as a regulatory mechanism, the underlying molecular details and consequences of this modification are not clearly articulated. Expanding on this point would help readers grasp the significance of pathway interplay in tumor progression.
In addition, the frequent use of abbreviations and specialized terms such as TET3, CBR1, ECM, and EMT without sufficient explanation may pose difficulties for readers who are less familiar with these concepts. Including brief definitions or context for these terms would make the section more accessible without sacrificing scientific rigor.
The therapeutic implications of TGF-β signaling are touched upon only briefly, with an emphasis on the potential benefits of dual pathway inhibition. However, the section would be strengthened by discussing current or emerging therapeutic approaches that specifically target TGF-β signaling in uterine tumors. This would provide a more translational perspective, linking molecular insights to clinical applications.
Finally, there are minor inconsistencies and vague statements that could be improved. For instance, the claim that TGF-β has dual roles in cancer is well-established, but the text could better tailor this information to the specific context of uterine tumors. Similarly, the observation that TGF-β3 expression is increased in ULM compared to myometrium would benefit from supporting quantitative data or citations to enhance credibility.
Overall, while the section effectively highlights important aspects of TGF-β signaling, addressing these issues—such as improving structural clarity, expanding on the relevance of ULMS, deepening mechanistic explanations, and elaborating on therapeutic prospects—would substantially enhance its clarity, balance, and impact.
Finally, the discussion section is excessively long and contains considerable redundancy. In my view, the information should be more concise and focused.
Author Response
Response to Reviewer 2:
We appreciate the reviewer's valuable comments and suggestions, which have helped improve the quality and clarity of our manuscript. Please find our point-by-point responses below.
- …In particular, the discussion on the potential progression from ULM to ULMS is repeated and somewhat disjointed. This could be streamlined by fully addressing the genomic controversy before transitioning to epigenetic mechanisms.
We appreciate this feedback and agree that the discussion was disjointed. We have streamlined this section to improve clarity and flow. However, given the manuscript's focus and length constraints, we have chosen to cover the genomic aspects only to the extent necessary for understanding the epigenomic mechanisms. We believe the revised structure now provides a more coherent transition from genomic context to epigenetic mechanisms while maintaining our manuscript's primary focus
- There’s also a sentence construction issue: “While genomic analyses have not definitively resolved this Emerging evidence has shown...” is grammatically incorrect and interrupts the flow. It should read: “While genomic analyses have not definitively resolved this, emerging evidence has shown...”
We thank the reviewer for this comment. We have carefully searched through our manuscript but were unable to locate the specific sentence construction mentioned ('While genomic analyses have not definitively resolved this Emerging evidence has shown...'). We have thoroughly reviewed our manuscript for similar grammatical issues and made corrections where needed. If the reviewer could provide a line number or section reference for this specific sentence, we would be happy to address it promptly.
- Additionally, the transitions between clinical background, molecular mechanisms, and the aims of the review are abrupt. Introducing brief linking sentences would improve readability. For example, before discussing epigenetics, a sentence like “This uncertainty has led researchers to explore epigenetic mechanisms as alternative diagnostic tools” would clarify the rationale.
Thank you for this feedback. We have revised the introduction to include linking sentences that better connect the clinical background, molecular mechanisms, and review aims. These transitions now create a clearer narrative flow from diagnostic challenges to epigenetic solutions, addressing the concern about abrupt transitions.
- Some sentences are overly long and could be split for clarity, especially in the paragraph on ULMS prognosis and treatment response. Finally, the objective of the review is implied but not explicitly stated concisely adding a clear closing sentence outlining the purpose (e.g., identifying epigenetic biomarkers and therapeutic targets) would strengthen the conclusion of the introduction.
Thank you for this feedback. We have shortened overly long sentences, particularly in the ULMS prognosis paragraph, and clarified the review objectives with a more concise statement of our aims to identify epigenetic biomarkers and therapeutic targets.
- I would recommend citing the most recent ESGO guidelines in the introduction, and placing greater emphasis on the preoperative diagnostic challenges associated with ULMS (e.g., PMID: 38054268; doi:10.1002/uog.20270)."
Thank you for this suggestion. We have incorporated the ESGO guidelines and enhanced the discussion of preoperative diagnostic challenges, including the referenced imaging studies.
- (Epigenetic alterations 2.1. DNA Methylation Landscapes) First, the narrative suffers from redundancy and lack of synthesis. Numerous studies are reported consecutively, often without critical integration or highlighting of overarching trends. For example, both Liu et al. and Miyata et al. are cited to support global hypomethylation in ULMS, but their conclusions are not clearly reconciled. The manuscript should focus less on listing study details and more on synthesizing their collective implications.
Thank you for this important feedback. We have restructured section 2.1 to better synthesize findings rather than listing studies consecutively. We now directly compare ULM and ULMS methylation enzyme profiles upfront and clearly reconcile apparent contradictions between studies. The revised text focuses on overarching trends and collective implications rather than individual study details.
- (Epigenetic alterations 2.1. DNA Methylation Landscapes) Sample size limitations are correctly acknowledged, but the text tends to overstate conclusions from underpowered studies. For instance, the statement that ULMS shows consistent CpG island hypermethylation is weakened by conflicting data (e.g., from Conconi et al.). The review should emphasize that current evidence is preliminary and sometimes contradictory, and should more strongly advocate for larger, well-controlled methylation studies.
We appreciate this critical point. The revised text now explicitly acknowledges that current evidence is preliminary and sometimes contradictory. We have removed overstated conclusions about CpG island patterns and now clearly present conflicting data from different studies. The text more strongly advocates for large-scale, well-controlled methylation studies as an urgent research priority.
- (Epigenetic alterations 2.1. DNA Methylation Landscapes) Language and sentence structure need tightening. Several sentences are overly long or packed with technical details that obscure the main point. For example:
Thank you for this valuable feedback. We have shortened sentences and separated complex constructions to improve reading flow, while removing non-essential information that did not contribute to the chapter's conclusions
- (Epigenetic alterations 2.1. DNA Methylation Landscapes) The section on hormone-related methylation changes is scientifically relevant but somewhat fragmented. The link between DNA methylation and hormone receptor signaling (PGR, ERα) is well-made, but the molecular mechanisms are described in too much detail without tying back clearly to clinical significance. It would help to summarize key messages—for example, that aberrant methylation in hormone-regulated pathways may explain differences in hormone responsiveness between ULM and ULMS.
We have restructured this section to better emphasize clinical significance. However, we believe readers need some molecular context to understand the bigger picture, so we maintained essential details while streamlining the section. We reduced excessive molecular detail and focused on how methylation patterns explain therapeutic responses, making clinical relevance more explicit.
- (Epigenetic alterations 2.1. DNA Methylation Landscapes) The final paragraph appropriately calls for larger studies, but it would be stronger if framed as a future research direction rather than just a knowledge gap. Instead of simply noting that ULMS is often studied as part of broader sarcoma cohorts, the authors could recommend design principles for future methylation studies that focus specifically on ULMS vs ULM.
We have reframed the conclusion as a proactive research agenda rather than simply noting knowledge gaps. The revised text now provides specific recommendations for future study design, advocating for large-scale, well-controlled methylation studies with clear design principles for ULMS vs ULM comparisons.
- (2.2 Histone Modifications and Chromatin Remodeling Mechanisms) First, the structure is overly dense and encyclopedic, with a succession of findings presented in rapid sequence and little attempt at integration or hierarchy. Important mechanisms—such as H3K27me3-mediated repression by EZH2, BRD9 involvement in chromatin remodeling, and ATRX loss triggering ALT—are discussed with limited connective analysis or commentary on their relative importance or potential interactions. The text would benefit from clearer transitions and grouping of findings into thematic subunits (e.g., “repressive modifications,” “hormone-epigenetic crosstalk,” “chromatin remodeler mutations”).
Thank you for the helpful feedback. To address this concern, we have restructured the section to provide a clearer framework for readers less familiar with epigenetic mechanisms. We now begin with a brief overview of key histone modifications and chromatin remodeling principles, followed by a thematically organized discussion that mirrors the order introduced. Specifically, we grouped the content into: (1) Histone methylation and acetylation—general concepts; (2) Chromatin remodeling complexes-general concepts; (3) Methylation patterns in ULM (limited data available for ULMS); (4) Acetylation-related changes in ULM and ULMS; (5) BRD9-associated chromatin remodeling; and (6) ATRX loss and the ALT phenotype.
- (2.2 Histone Modifications and Chromatin Remodeling Mechanisms) Redundancy and excessive detail also limit readability. Some sentences provide mechanistic minutiae that obscure the main point. For example, mentioning the precise phosphorylation mechanism of EZH2 via PI3K/AKT in the context of ER signaling may be more appropriate for a specialized mechanistic discussion than a broad tumor biology section. Similarly, the detailed breakdown of H2A.Z alterations in the rare SRCAP-mutated leiomyoma subtype could be abbreviated or moved to a footnote/reference, given its limited prevalence (~1%).
Given the overall length of the article and the additional introductory paragraphs we incorporated to improve clarity, we agree that streamlining is essential for readability. As such, we have omitted the mechanistic detail regarding EZH2 phosphorylation via PI3K/AKT and the description of H2A.Z alterations in SRCAP-mutated leiomyomas
- (2.2 Histone Modifications and Chromatin Remodeling Mechanisms) The discussion of ULM vs ULMS differences is crucial but not sufficiently emphasized or critically interpreted. While ATRX loss and ALT activation in ULMS are well described, the implications of these changes for tumor behavior and prognosis deserve stronger synthesis. For example, a comparative sentence summarizing how histone and chromatin remodeling alterations contribute to the benign vs malignant phenotypes would help contextualize the described findings.
We added a sentence contextualizing the findings. How ATRX loss can serve as a diagnostic and prognostic biomarker, and how precisely it can be targeted (PARP inhibitors, WEE1 inhibitors, and G-quadruplex stabilizers)
- (2.2 Histone Modifications and Chromatin Remodeling Mechanisms) The role of therapeutic targets (e.g., HDAC inhibitors, BRD9 inhibitors) is promising, but the therapeutic relevance is dispersed across the section without a conclusive synthesis. It would be helpful to highlight these mechanisms in a final paragraph that frames them as candidate targets, distinguishing those with preclinical validation from more speculative ones.
We revised the text to more clearly highlight the therapeutic relevance of the discussed mechanisms. Specifically, we now emphasize that BRD9 inhibition with TP-472 has demonstrated preclinical efficacy, distinguishing it as a more validated candidate compared to other, more speculative targets.
- (2.2 Histone Modifications and Chromatin Remodeling Mechanisms) Finally, while the language is generally precise, sentence structure can be overly long and passive, reducing clarity. Rewriting complex ideas into shorter, active sentences would improve accessibility for readers less familiar with epigenetic terminology.
We have rewritten many sections to improve readability.
- (2.3.1. ULM-specific miRNA patterns) This section addresses an emerging and relevant topic in USMT biology. The role of miRNAs and lncRNAs is well articulated, but the presentation is fragmented and could benefit from thematic organization. Rather than listing individual miRNAs and lncRNAs, grouping them by functional axis—e.g., ECM regulation (miR-29, H19), hormonal signaling (miR-206, miR-27b, XIST), or Wnt/β-catenin pathway (LINCMD1)—would enhance clarity and biological interpretation.
We have reorganized the ULM miRNA section with thematic grouping by functional pathways to improve clarity and biological interpretation as well as added a separate lncRNA section.
- (2.3.1. ULM-specific miRNA patterns) The discussion on miR-29 and XIST is particularly strong, though somewhat repetitive. The interplay between hormones, miRNA expression, and downstream signaling is compelling but would be clearer with more concise language and a brief integrative summary.
The miR-29 and XIST discussion has been streamlined to reduce repetition while maintaining the key mechanistic insights.
- (2.3.1. ULM-specific miRNA patterns) The section lacks a direct comparison between ULM and ULMS non-coding RNA profiles, which would help contextualize miRNA roles in benign vs malignant progression. Mention of diagnostic potential (e.g., liquid biopsy) is promising but underdeveloped.
We have added sections 2.3.5 and 2.3.6 that directly compare ULM and ULMS miRNA profiles, highlighting both shared patterns (let-7, miR-200 families) and divergent signatures (miR-221, miR-206) to contextualize benign versus malignant characteristics. A summary table of key miRNA expression profiles has been included to facilitate clinical interpretation.
- (3.2 ULMS-specific miRNA patterns) However, the mechanistic insights remain mostly inferred from other cancers, reflecting a gap in direct ULMS-specific functional studies. This limits the strength of conclusions about causality and therapeutic implications. The mention of circulating miRNAs and their impact on treatment response is promising but brief; expanding on clinical validation status would improve translational relevance.
We have explicitly acknowledged that mechanistic insights in ULMS remain largely inferred from other cancer types and clearly identified the lack of ULMS-specific functional studies as a significant knowledge gap. The subject of circulation miRNA is covered in the biomarker chapter.
- (3.1 Wnt/β-Catenin Pathway) Certain aspects could be further refined for clarity and depth. For instance, the description of epigenetic changes, such as the reduction in H3K4me3 and its downstream effects on SATB2 and SHOX2, is somewhat brief. Expanding on how these chromatin modifications mechanistically promote Wnt signaling would strengthen the reader’s understanding of the pathway’s regulation.
We have expanded the discussion on the downstream effects of SATB2 and SHOX2. However, the precise mechanisms by which these factors promote Wnt/β-catenin signaling remain incompletely understood. The available primary data largely demonstrate correlative associations between SATB2 and SHOX2 expression and Wnt pathway activation, without fully elucidating the underlying molecular mechanisms. In the case of SHOX2, the upregulation of RUNX2—a known activator of Wnt/β-catenin signaling—likely serves as a key intermediary. For SATB2, we have elaborated on its function as a chromatin organizer that modulates higher-order chromatin structure, thereby facilitating transcription of Wnt/β-catenin pathway genes. Nonetheless, the exact regulatory cascade linking SATB2 activity to canonical Wnt signaling remains to be fully defined
- (3.1 Wnt/β-Catenin Pathway) Likewise, the role of non-canonical Wnt signaling receptors like ROR1 in ULMS is introduced but not fully developed. Providing additional context or evidence on how reduced ROR1 expression might shift signaling towards canonical β-catenin activation would enrich the discussion.
We have expanded the discussion on ROR1 as suggested. ROR1 is a pseudokinase structurally related to ROR2, for which the functional data are somewhat more robust. Both receptors are known to bind non-canonical Wnt ligands, particularly Wnt5a. While the precise mechanisms remain incompletely understood, current evidence suggests that ROR1 can modulate canonical Wnt/β-catenin signaling indirectly through activation of non-canonical pathways such as Wnt/Ca². In particular, Wnt5a–ROR1 interaction has been associated with antagonism of canonical Wnt signaling, potentially through mechanisms involving competition for Wnt ligands or interference with canonical receptor complex formation. Although these mechanisms are frequently cited, it is important to note that they remain hypotheses, and we were unable to identify definitive experimental studies that conclusively demonstrate these mechanisms. We have clarified this in the revised manuscript and cited representative literature accordingly.
- (3.1 Wnt/β-Catenin Pathway) The therapeutic implications section is a valuable inclusion, effectively tying the mechanistic insights to potential clinical strategies. Nevertheless, it would benefit from a more detailed consideration of challenges involved in targeting these pathways, such as the complexity of pathway crosstalk or potential resistance mechanisms, to provide a more balanced perspective.
To maintain the logical flow of the manuscript, this content was incorporated into Section 5, which addresses the broader therapeutic potential. In this section, we provide a more detailed consideration of the challenges associated with targeting the Wnt/β-catenin pathway. Specifically, we highlight that β-catenin plays essential roles in development, tissue maintenance, and homeostasis, which significantly restricts the therapeutic window. Additionally, we discuss the extensive crosstalk between canonical and non-canonical Wnt signaling, as well as interactions with other pathways such as Notch and TGF-β, which can contribute to compensatory resistance mechanisms.
- (3.2 PI3K/AKT/mTOR Pathway) The section contains some lengthy and complex sentences that might hinder reader comprehension. Breaking these into shorter, clearer sentences and perhaps using subheadings or bullet points to separate mechanisms in ULM and ULMS could improve readability.
Thank you for this feedback. We restructured the section with clear subheadings (3.2.1-3.2.6) and broke down complex sentences to improve readability.
- (3.2 PI3K/AKT/mTOR Pathway) Additionally, referring to Figure 1 multiple times without summarizing its key elements assumes prior familiarity and could leave some readers behind; a brief description in the text would be helpful.
Thank you for this suggestion. We added a comprehensive description of Figure 1 and 2 in each section explaining its layout, color coding system, and key regulatory elements to help readers interpret the figure references throughout the text.
- (3.2 PI3K/AKT/mTOR Pathway) Minor technical points include ensuring consistent abbreviation definitions (e.g., spelling out “gain-of-function” at first mention of MED12 mutation) and cleaning up residual editorial notations present in the text.
Thank you for pointing this out. We have reviewed the entire document to ensure consistent abbreviation definitions at first mention and eliminate any residual editorial notations. While we have systematically defined all abbreviations upon initial use, we have prioritized defining those most essential for manuscript comprehension to maintain readability and focus on the core scientific content
- (3.2 PI3K/AKT/mTOR Pathway) However, the complex and sometimes contradictory effects of progesterone receptor signaling on tumor growth could be discussed in more detail to clarify this important aspect.
Thank you for this important comment. We have expanded the discussion to clarify the mechanistic basis of progesterone's complex effects, including rapid non-genomic PI3K/Akt pathway activation and experimental evidence of receptor-dependent signaling. This could explain why progesterone receptor inhibition reduces bleeding symptoms but inconsistently affects tumor volume.
- (3.2 PI3K/AKT/mTOR Pathway) While the section is rich in detail and well referenced, the text could benefit from improved structural organization. Currently, many concepts are presented densely within single paragraphs, which might challenge readers less familiar with the topic. Dividing the content into clearly delineated subsections or thematic paragraphs (e.g., genomic alterations, epigenetic regulation, microRNA involvement, hormonal regulation) would enhance readability and comprehension
Thank you for this valuable feedback on improving the structural organization of section 3.2. As addressed in our response to comment 23, we have reorganized this section with clear subheadings to separate thematic content areas, which directly addresses your concern about dense presentation within single paragraphs and enhances overall readability.
- (TGF-β signaling) …the text could benefit from a clearer structure to improve readability and flow. Currently, the information is densely packed, which makes it challenging to follow the logical progression of ideas, especially for readers who may not be experts in this signaling pathway. Organizing the content into more distinct thematic paragraphs—such as an introduction to the pathway, epigenetic regulation, hormonal influences, and therapeutic implications—would enhance clarity and engagement.
Thank you for this valuable feedback. We have substantially reorganized the section to improve logical flow and readability. The content is now structured thematically, beginning with an introduction to the pathway and its interactions with PI3K/Akt, followed by its specific roles in ULM, and concluding with its emerging importance in ULMS. This reorganization creates a clearer narrative progression while maintaining scientific rigor. Given the moderate scope of TGF-β research in uterine tumors compared to other pathways, we believe this restructured format provides optimal clarity without requiring additional subheadings.
- (TGF-β signaling) Furthermore, while the section recognizes that TGF-β is less studied in ULMS compared to ULM, the discussion around ULMS remains brief and underdeveloped. The example of CBR1's role in modulating TGF-β signaling in ULMS is interesting but deserves a deeper exploration to provide a fuller understanding of how TGF-β contributes to ULMS pathobiology. This imbalance between ULM and ULMS content limits the comprehensiveness of the review.
We have significantly expanded the ULMS discussion, particularly the CBR1-TGF-β relationship. The revised text now provides detailed mechanistic insights into how CBR1 disrupts autocrine TGF-β signaling loops in ULMS cells, with corresponding figure updates. This expansion addresses the content imbalance while acknowledging current research limitations in ULMS TGF-β signaling
- (TGF-β signaling) The explanation of mechanistic interactions, particularly the crosstalk between TGF-β and PI3K/Akt pathways, is somewhat superficial. Although Akt phosphorylation of SMAD proteins is mentioned as a regulatory mechanism, the underlying molecular details and consequences of this modification are not clearly articulated. Expanding on this point would help readers grasp the significance of pathway interplay in tumor progression.
We appreciate this feedback and have enhanced the mechanistic explanations. However, we have balanced mechanistic depth with accessibility for our target audience. While more detailed molecular interactions could be described, we believe the current level provides sufficient understanding of pathway significance in ULM and ULMS without overwhelming readers with excessive biochemical complexity that may not directly advance clinical understanding
- (TGF-β signaling) In addition, the frequent use of abbreviations and specialized terms such as TET3, CBR1, ECM, and EMT without sufficient explanation may pose difficulties for readers who are less familiar with these concepts. Including brief definitions or context for these terms would make the section more accessible without sacrificing scientific rigor.
Thank you for highlighting this concern. We have conducted a systematic review of abbreviations throughout the entire manuscript, ensuring proper definition at first mention and organizing the abbreviations list alphabetically according to journal requirements. We have also reduced abbreviation density in individual paragraphs to improve reading flow. While specialized terminology remains necessary for scientific precision, we believe these modifications enhance accessibility while maintaining clarity for our target audience.
- (TGF-β signaling) The therapeutic implications of TGF-β signaling are touched upon only briefly, with an emphasis on the potential benefits of dual pathway inhibition. However, the section would be strengthened by discussing current or emerging therapeutic approaches that specifically target TGF-β signaling in uterine tumors. This would provide a more translational perspective, linking molecular insights to clinical applications.
This observation is accurate, and we have addressed therapeutic implications in our expanded clinical applications section rather than within this mechanistic overview. Our manuscript focuses primarily on epigenomic similarities and differences between ULM and ULMS, with pathway convergence representing a novel mechanistic insight. While pathway inhibitor studies strengthen the importance of these molecular targets, detailed therapeutic status discussions would extend beyond our current scope and warrant dedicated review articles. However, we have expanded the clinical section to provide readers with essential translational perspectives linking these molecular insights to therapeutic applications.
- (TGF-β signaling) Finally, there are minor inconsistencies and vague statements that could be improved. For instance, the claim that TGF-β has dual roles in cancer is well-established, but the text could better tailor this information to the specific context of uterine tumors.
We have thoroughly revised the text to eliminate inconsistencies and clarify previously vague statements. Our literature review identified limited evidence specifically addressing TGF-β's dual role in uterine muscular tumors. While additional studies exist for endometrial or cervical carcinomas, we avoided incorporating findings from these different cellular origins to maintain focus on leiomyoma/leiomyosarcoma-specific mechanisms and prevent scope expansion beyond our manuscript's objectives.
- (TGF-β signaling) Similarly, the observation that TGF-β3 expression is increased in ULM compared to myometrium would benefit from supporting quantitative data or citations to enhance credibility.
Unfortunately, the referenced studies did not provide specific quantitative data for TGF-β3 upregulation levels in ULM compared to normal myometrium. We acknowledge this limitation while maintaining the qualitative finding supported by the available literature. Future studies with quantitative expression analyses would strengthen this observation.
- (TGF-β signaling) Overall, while the section effectively highlights important aspects of TGF-β signaling, addressing these issues—such as improving structural clarity, expanding on the relevance of ULMS, deepening mechanistic explanations, and elaborating on therapeutic prospects—would substantially enhance its clarity, balance, and impact.
We have comprehensively addressed the structural reorganization, ULMS content expansion, and mechanistic detail enhancement as requested. The CBR1-ULMS relationship has been substantially developed with corresponding figure modifications. Therapeutic aspects receive appropriate attention in our clinical applications section, maintaining manuscript coherence while providing translational insights. These revisions significantly improve section clarity, balance, and scientific impact while preserving focus on our primary research objectives.
- (Discussion) Finally, the discussion section is excessively long and contains considerable redundancy. In my view, the information should be more concise and focused.
We appreciate this feedback and have extensively revised our future directions section. We agree that a big part was repeating information, we now focused more on the take home messages and are highlighting the most prominent gaps in current research.
We believe these revisions have significantly strengthened the manuscript and addressed all the reviewer's concerns comprehensively. We are grateful for the opportunity to improve our work and look forward to your further evaluation.
Reviewer 3 Report
Comments and Suggestions for Authors
The manuscript by Pfaff et al., titled “Decoding the Epigenome: Comparative Analysis of Uterine Leiomyosarcoma and Leiomyoma,” provides a review of the epigenetic similarities and differences between these two tumor types, aiming to identify diagnostic biomarkers and potential therapeutic targets that could enhance clinical management. Overall, the article offers a valuable overview of the field; however, several aspects require improvement to strengthen its scientific clarity and comprehensiveness.
Major comments
- In Table 1, please include an additional column indicating the sample types used in each study, such as tissues, cells (specifying whether primary or cell lines), or both.
- Similarly, in Table 2, please add a column describing the cell types (primary vs. cell lines) as well as the in vivo model, if applicable.
- The discussion on the interplay between key signaling pathways and the epigenome remains vague. In addition to the current section addressing histone modifications (Lines 517–523), the authors should also include a discussion on targeting DNA methylation in both tumor types. This addition should be supported with appropriate references and aligned with available relevant figure(s), to provide a more comprehensive overview of epigenetic therapeutic strategies.
- The "Future Directions" section lacks key elements that would strengthen its perspective on areas requiring further investigation. In particular, the regulatory and functional roles of specific epigenetic mechanisms in both benign (ULM) and malignant (ULMS) uterine tumors are not adequately addressed. For example, comparative spatial transcriptomics integrated with epigenetic profiling is needed to reveal tumor heterogeneity and the influence of the tumor microenvironment between ULM and ULMS in the future study. Additionally, RNA methylation pathways, such as N6-methyladenosine (m⁶A) modification, and the role of BET (bromodomain and extra-terminal domain) proteins, which have been studied in ULMS, warrant focused investigation in ULM as well. These studies may uncover key differences in epigenetic regulation and help in identifying novel therapeutic targets.
Minor Comments
- In Table 2, please correct “EZH2 inhibitor” to “EZH2 inhibitors.”
- Ensure that all abbreviations are defined upon first use, for example, EMT (line 561) and ECM (line 571).
- Please italicize all gene names throughout the manuscript, such as TET1 and TET3 (line 149).
Author Response
Response to Reviewer 3:
We appreciate the reviewer's valuable comments and suggestions, which have helped improve the quality and clarity of our manuscript. Please find our point-by-point responses below.
- In Table 1, please include an additional column indicating the sample types used in each study, such as tissues, cells (specifying whether primary or cell lines), or both
For Table 1, all methylation data from the cited references were investigated exclusively on tissue samples. Rather than adding a repetitive column, we included this information in the table caption to maintain better readability and avoid redundancy
- Similarly, in Table 2, please add a column describing the cell types (primary vs. cell lines) as well as the in vivo model, if applicable.
As recommended, we have included information about tested sample types (primary vs. cell lines) and experimental models (in vivo/in vitro) in an additional column in Table 3. For completeness, we applied the same approach to the new Table 2 and Table 4, ensuring comprehensive documentation of experimental conditions across all tables.
- The discussion on the interplay between key signaling pathways and the epigenome remains vague. In addition to the current section addressing histone modifications (Lines 517–523), the authors should also include a discussion on targeting DNA methylation in both tumor types. This addition should be supported with appropriate references and aligned with available relevant figure(s), to provide a more comprehensive overview of epigenetic therapeutic strategies.
We've broadened our discussion on DNA methylation and various pathways, especially WNT-beta catenin, where the evidence is quite strong. We also conducted a second literature search, and unfortunately, the current research only supports the points we've already included. If there are any specific details you'd like us to add that we might have overlooked in our initial and follow-up reviews, we'd be happy to include them.
- The "Future Directions" section lacks key elements that would strengthen its perspective on areas requiring further investigation. In particular, the regulatory and functional roles of specific epigenetic mechanisms in both benign (ULM) and malignant (ULMS) uterine tumors are not adequately addressed. For example, comparative spatial transcriptomics integrated with epigenetic profiling is needed to reveal tumor heterogeneity and the influence of the tumor microenvironment between ULM and ULMS in the future study. Additionally, RNA methylation pathways, such as N6-methyladenosine (m⁶A) modification, and the role of BET (bromodomain and extra-terminal domain) proteins, which have been studied in ULMS, warrant focused investigation in ULM as well. These studies may uncover key differences in epigenetic regulation and help in identifying novel therapeutic targets
We thank the reviewer for this valuable comment. In alignment with the suggestion from Reviewer 2 regarding the need for a more concise and focused discussion, we have comprehensively revised the "Future Directions" section. While the regulatory and functional roles of epigenetic mechanisms were already discussed in detail throughout the main sections of the manuscript, we now more clearly summarize their implications for future research directions in the revised section.
We appreciate the emphasis on integrative approaches such as spatial transcriptomics combined with epigenetic profiling to better understand tumor heterogeneity and the role of the microenvironment. This perspective has been incorporated, and we now propose specific study designs that address these aspects. Regarding BET proteins, we have already discussed BRD9 inhibition as a promising therapeutic approach in our manuscript.
We believe that the revised section now presents a more structured and actionable research roadmap, highlighting specific epigenetic targets and methodological approaches that can guide future investigations while focusing on areas with sufficient evidence base to support meaningful clinical translation.
- Minor comments: In Table 2, please correct “EZH2 inhibitor” to “EZH2 inhibitors.“; Ensure that all abbreviations are defined upon first use, for example, EMT (line 561) and ECM (line 571).; Please italicize all gene names throughout the manuscript, such as TET1 and TET3 (line 149)
"We appreciate these detailed suggestions. We have corrected "EZH2 inhibitor" to "EZH2 inhibitors" in Table 2 and ensured all abbreviations are defined upon first use, particularly for key terms central to our discussion. We have maintained standard formatting conventions by italicizing gene names while keeping protein names in regular font.
We believe these revisions have significantly strengthened the manuscript and addressed all the reviewer's concerns comprehensively. We are grateful for the opportunity to improve our work and look forward to your further evaluation.

Round 2
Reviewer 1 Report
Comments and Suggestions for Authors
Thank you for correcting the manuscript. A few minor issues left as follows:
- Table 1 - something wrong with 3rd column.
- Several dots were lost while correcting the manuscript.
Author Response
Dear Reviewer,
Thank you so much for your comments. We carefully reviewed Table 1 but were unable to identify a specific misalignment. However, we did find a misalignment in Table 3, which has been corrected. We will ensure a thorough review during the proofing stage to maintain proper alignment and clarity. We also conducted a comprehensive review of the entire document and identified one missing period, which has been added:
Line 652: Inserted a period after "LSC" in "LSC. The function..."
Additionally, we corrected several minor spacing inconsistencies that did not affect readability but have now been standardized:
-
Removed unnecessary spaces before periods (Lines 202, 304, 318, 327, 330, 353, 358, 379, 598)
-
Reduced excessive spacing at the beginning of sentences (Lines 581, 611)
I am confident that these revisions address the formatting concerns.
Reviewer 2 Report
Comments and Suggestions for Authors
"In my opinion, the authors have significantly improved the manuscript compared to the first version
Author Response
We appreciate the positive feedback and are glad that the improvements were well received.
Reviewer 3 Report
Comments and Suggestions for Authors
I agree to publish this review manuscript.
Author Response
We appreciate your positive evaluation and support for publication.